# Application of quantile regression to examine changes in the distribution of Height for Age (HAZ) of Indian children aged 0–36 months using four rounds of NFHS data

**Thirupathi Reddy Mokalla**[1], **Vishnu Vardhana Rao Mendu**[2]*

**1** Biostatistics, Public Health Nutrition Division, ICMR-National institute of Nutrition, Jamai-Osmania, Hyderabad, Telangana, India, **2** Department of Health Research, ICMR-National Institute of Medical Statistics, MOHFW, New Delhi, India

* Dr_vishnurao@yahoo.com

## Abstract

### Background

The prevalence of stunting among under- three Indian children though decreased, still it is considered to be alarmingly high. In most of the previous studies, traditional (linear and logistic) regression analyses were applied. They were limited to encapsulated cross-distribution variations. The objective of the current study was to examine how the different determinants were heterogeneous in various percentiles of height for age (HAZ) distribution.

### Methods and findings

This article examined the change in the HAZ distribution of children and examined the relationships between the key co-variate trends and patterns in HAZ among children aged <3 years over a period of 24 years. Four successive rounds of the National Family Health Survey data 1992–93, 1998–99, 2005–06, and 2015–16 were used for analysis. The final study included 206579 children aged <3 years (N = 106136 male, 100443 female). To explain and analyse differences in the HAZ distribution, the lambda-mu-sigma (LMS) method was used. Trends in height for age (HAZ) distribution over time were analysed using separate gender-stratified quantile regression (QR). The selected socio-economic, demographic and other predictors were considered for this analysis. The quantile regressions have indicated that mothers who have higher than primary level education were more proactive in mitigating malnutrition among children at the lower end of the distribution. The age, birth order, mother's body-mass-index (BMI) and economic status, among children, were some more determining factors for HAZ. Results of selected quantile regression were estimated at the 5th, 10th, 25th, 50th, 75th, 90th, and 95th quantiles.

### Conclusions

The outcome of various covariates working differently across the HAZ distribution was suggested by quantile regression. The major discrepancies in different aspects were underlined

**Data Availability Statement:** All the supporting data are available in the manuscript. We used secondary data "NFHS- 1,2,3 and 4" for this study,

which is publicly accessible at https://dhsprogram.
com/what-wedo/survey/survey-display-355.cfm.

**Funding:** The author(s) received no specific
funding for this work.

**Competing interests:** The authors have declared
that no competing interests exist.

by socioeconomic and demographic aspects among the Indian population. The heterogeneity of this effect was shown using quantile regression. Policymakers may choose to concentrate on the most important factors when formulating policies to lessen the prevalence of stunting in India.

## Introduction

In India, malnutrition has always been a significant public health problem among children. The survival and early development of children have been shown to be severely affected by malnutrition. In addition, it vigorously affects the health of pregnant and nursing mothers. The overall resistance-to diseases and future performance in school and at work was also associated with Malnutrition [1, 2]. In this study we explore the risk factors for childhood malnutrition in India, which is one of world's fastest growing economies, and the second most populated country in the world [3, 4]. Nutrition has been an important role in determining the health status, particularly in developing nations [5].

The term malnutrition, which includes both undernutrition and overnutrition, is a significant causative factor for child mortality worldwide [6]. This research was based on the whole Height for age-Z values (HAZ) distribution using the Quantile Regression Technique (QR). It may assist to identify the variables that influence HAZ by dividing the distribution into distinct quantiles instead of the population average score [7]. The three characteristics describing the physical growth of children are stunting (short height for age), wasting (low weight for height/ length) and underweight (low weight for age) [8–10]. According to the National Family Health Survey (NFHS-1,2,3 and 4), the prevalence of stunting among the children aged 0–36 months was 44.6%, 45.5%, 41.4%, and 36.6%, respectively [11–14]. Stunting and other forms of undernutrition reduces a child's chance of survival [15]. It was characterised as the percentage of children aged 0 to 36 months whose height for age was less than two standard deviations (moderate stunting) and less than minus three standard deviations (severe stunting) from the median of the WHO Child Growth Standards [16, 17]. We have applied HAZ for assessing the growth at a given point in time and also allowing comparisons between gender and age groups.

In early childhood, particularly, stunting increases the chance of susceptibility to sickness. Since stunting is associated with suboptimal brain development, it has practically permanent effects on physical, mental development and would have long- lasting adverse ramifications on the intellectual capacity, school performance, and future benefits of the children [15]. The effects mentioned above on development and uniformity caused by child stunting are most likely to have the outcome of keeping children at the midpoint of poverty, which in turn adversely affects the development of nations [18, 19]. Thus, improving children's nutritional status and lessening the prevalence of malnutrition to advance their physical and mental improvement would be extraordinarily significant [20]. Better wellbeing and nourishment result in financial development [21, 22]. According to the WHO, stunting is a reliable measure of overall social deprivation [23].

Child health is predominantly noteworthy because of its connection to child poverty and the build-up of adult human capital. The improvement of the needy children's health and nutrition has been an effective method of improving school participation and upgrading the financial status since learning converts into gains through long-run productivity [24, 25].

There has been a considerable literature analysing the determinants of nourishing child status in a broad scope of nations utilising distinctive econometric techniques. The earlier studies

have used standard multiple linear or logistic regression models. The more significant part of those past investigations was that their emphasis was on finding the determinants at the mean and odds using logistic regression. Traditional regression modelling approaches (linear/logistic regression) used to estimate the risk factors of childhood stunting often tend to oversimplify the complex interplay of a battery of risk factors through emphasizing a solitary risk and intervention [9]. Recent studies in public health and epidemiological research called upon the application of a systems approach that focuses on examining the determinants of stunting in its entirety and seek to avoid the possibility of incorrect estimation of risk factors and programmatic interventions owing to any oversimplification. In addition, recent studies have also noted that the impact of risk factors at the lower quantile of the distribution could be considerably different contrary to the population mean or at the higher end of the distribution. Therefore, in any population with a considerable degree of nutritional transition and disproportionate burden of anthropometric failure, it is critical to recognize and address the risk factors of stunting at the different ends of the spectrum and meticulously identify the actual effect of the risk factors at a different parts of the quantile distribution) to design and implement appropriate interventions. Quantile regression model quantiles of the outcome as a function of covariates and provides an opportunity to examine whether covariates have differential effects across the height-for-age z-score distribution, particularly towards the lower quantile. On the other hand, previous studies have mostly examined the determinants of stunting by applying logistic regression models for dichotomized versions of the Z-score (e.g., stunted vs. not stunted) or linear regression models for the continuous Z-score [7, 26–34].

The utilisation of these methods may prompt inaccurate strategy intercession measures if the relationship between child nutritional status and the specific socioeconomic and demographic determinants is heterogeneous at the various percentiles of the nutritional distribution. Only a few studies have focused on the distinction at various points of the conditional nutritional dispersion [28, 34, 35].

The limitation of previous studies can be resolved by analysing demographic and socioeconomic factors across varied points of the conditional HAZ distribution in India. Moreover, the changes in the stunting distribution of children and connections between the key covariates' patterns and trends in stunting were investigated by using the quantile regression model [36].

The objective of this paper was to find the association between demographic, socioeconomic and health factors of child nutritional status from 4 rounds of National Family Health Survey (NFHS) data of children under 3 years of age years in India over 24 years by using quantile regression. In our opinion, adequate research has not been attempted in this area. The current analysis expects to improve the structure of successful intervention measures designed to tackle child malnutrition and improve child health.

## Materials and methods

### Data

Data for the analysis were taken from four consecutive rounds of the NFHS of India conducted during 1992–93, 1998–99, 2005–06, and 2015–2016. The NFHS is a large-scale community-based survey, carried out at household level, in the states and the Union territories of India. The International Institute for Population Sciences (IIPS), Mumbai, India, directed the numerous rounds of the survey with community-oriented assistance from a few national and international associations [12]. The surveys conducted led to more information on reliable estimates of fertility, Infant and children mortality, nutritional status of children, better use of MCH services at a national, state-level and across the urban and rural residence. The initial three rounds of the NFHS were intended to give state-level information. Nonetheless, the

fourth round of study, which includes a more prominent sample size, yields assessments of most factors for all 640 districts in India [14]. Each of the four rounds of the study incorporated multi-stage sampling design, two-stage sampling design in rural areas, and three stages in urban areas [37]. The NFHS gathered information utilising distinctive interview schedules, household schedules and qualified women/individual schedules, and for the fourth round, the Biomarker Questionnaire was incorporated. In all the rounds, the substance of the schedule remains the same [11–14]. This study used publicly accessible data from four rounds of National Family Health Survey (NFHS). Ethical approval was not needed for this study as NFHS obtained ethical clearance for their survey.

The household response rate of the first round was 96% and was 98% in the remaining three surveys. The individual response rate for the first, second, third and fourth rounds was 96%, 96%, 94% and 97% respectively [11–14]. All four rounds of the NFHS furnish data on anthropometric pointers with a different age group of children. For instance, NFHS-1 gathered data from children aged below four years and NFHS-2 gathered data from children below three years, while NFHS-3 and 4 gathered data from children below five years [11–14]. Hence, to make the assessments equivalent, the examination was limited to children age indicators of unit-level data of 206579 children aged <3 years (N = 106136 male and 100443 female) over 24 years period were considered. Statistically significant values $p < 0.01$, $p < 0.05$, and $p < 0.1$ were considered.

## Dependent variable

In this analysis, only HAZ was used as a dependent variable to find the independent variable's distribution on different quantiles. To estimate the HAZ marker, the reference population of WHO 2006 was considered [38]. As per the WHO, the child is classified as stunted if the height-for-age Z-score is < -2 standard deviations (SDs). If the child's Z score value is < - 3, it was classified as severely stunted. The HAZ score value considered from -6 to +6 remaining cases were excluded/flagged.

## Independent variable

This article focuses on the discussion and the identification of stunting in low- and middle-income countries as a major public health issue. In India, according to the past research, socioeconomic (maternal education, poverty) and biological variables (age, birth weight and maternal BMI) have been proven to have an effect on a child's nutritional health [1, 3, 24, 26, 29]. Five quantiles of the socioeconomic variable wealth index have been taken into account in this study and the quantiles are (poorest, poorer, middle, richer, and richest). Demographic variables such as caste—Scheduled Caste (SC), Scheduled Tribe (ST), Other Backward Class (OBC) and other castes were taken into consideration. Child characteristics such as age in months (0–6, 7–12, 13–24 and 25–59), birth order (1, 2, 3, 4 and above), and size at the time of birth (small, average and large) were taken into consideration. The maternal factors considered were maternal age of 15–49 years, BMI of mothers (underweight: Body mass index <18.5, normal: 18.5<BMI<25, and overweight BMI>25.0), and mother's education (No education, <5 years of schooling, 5–7, 8–9, 10–11, 12 years or more). Since wealth index data were not present in the first two rounds of NFHS, it was estimated only for the third and fourth surveys. Furthermore, the BMI (maternal) predictor data was not present in the NFHS1 survey; however, it was evaluated for the NFHS 2, 3 and 4 surveys.

## Statistical analysis

**Quantile regression.** Koenker and Bassett (1978) first introduced the key idea of quantile regression [7, 18, 39–41]. This procedure has an advantage over the conventional common least-squares method. It does not accept a steady effect of the independent factors over the

dependent variable's whole distribution [36, 39]. This methodology was utilised to consider a heterogeneous impact of every determinant alongside various percentiles of the dependent variable's conditional distribution [37].

Koenker and Bassett (1978) show that the empirical quantile function is the solution of the minimisation problem defined by [40]:

$$\hat{\beta}_\tau = \underset{\beta_\tau \in R^K}{argmin} \left\{ \sum_{i:y_i \geq x_i'\beta_r} \tau|y_i - x_i'\beta_\tau| + \sum_{i:y_i < x_i'\beta_r} (1-\tau)|y_i - x_i'\beta_\tau| \right\}$$

$$= \underset{\beta_\tau \in R^K}{argmin} \sum_i \rho_\tau|y_i - x_i'\beta_\tau|$$

With $\rho_\tau(z)$ can be defined as:

$$\rho_\tau(z) = \begin{cases} \tau(z) & \text{if } z \geq 0 \\ (\tau-1)z & \text{if } z < 0 \end{cases} = (\tau - I(z<0))z$$

Let $x_i$ where i = 1, ..., n a sample, a $K \times 1$ vector of regressors, $y_i = x_i'\beta_\tau + \varepsilon_{\tau_i}$, $0 < \tau < 1$, $\rho_\tau(z)$ is the check function, and I(•) the usual indicator function.

In this study, quantile regression analysis was performed to identify the independent variables related to child anthropometric Z score over the seven (5th, 10th, 25th, 50th, 75th, 90th, and 95th) percentiles. This analysis was performed using the SAS University Edition software and SAS® OnDemand for Academics.

**Lambda Mu Sigma (LMS) method.** Using quantile regression, this study investigated longitudinal changes in the HAZ distribution overtime for the separate gender-stratified. In addition, we used the LMS method to determine the age-specific secular trends in the child nutritional status measures, thereby allowing us to examine the temporal trends in selected seven percentile points of the stunting.

The LMS method summarises the shifting distribution by three curves representing the median, coefficient of variation and skewness; the latter expressed as a Box-Cox power transformation [42–44]. For a given value of covariate, to transform the response to standard normality LMS technique applies a Box-Cox transformation. In order to get the quantiles, an inverse Box- Cox transformation was applied to the quantiles of the standard normal distribution [45, 46]. A large sample size is required to gauge the percentiles in each age group with sufficient accuracy. The division may lose data from nearby groups. To maintain a strategic distance from division, Cole and Green (1992) proposed a Box-Cox transformation based on the semiparametric method from the LMS technique presented by Cole in 1988 [47].

We utilised the LMS technique in the GAMLSS package R version 3.4.3 (R Development Core Team, Vienna, Austria) to obtain figures of stunting distribution for 1992 and 2016, along with the lines showing the adjustments in the stunting measures [48, 49]. Additionally, we also utilised the LMS method to decide the age-specific patterns in the HAZ measures for both girls and boys, along with the lines permitting us to analyse the worldly patterns in specific percentiles of HAZ.

## Results

Table 1 presents descriptive statistics for the individual demographic, socioeconomic and health-related variables for each gender.

The calculated coefficients for seven percentiles (5th, 10th, 25th, 50th, 75th, 90th, and 95th) of males and females were calculated separately for the NFHS 1, 2, 3 and 4 are presented in Tables 2–5.

**Table 1. Descriptive statistics and prevalence of stunting children aged 0–36 months by factors at different categories based on (NFHS- 1, 2, 3, 4) data.**

| Variable (n and %) in each category) | Category | NFHS-1 | | NFHS-2 | | NFHS-3 | | NFHS-4 | |
|---|---|---|---|---|---|---|---|---|---|
| | | Female | Male | Female | Male | Female | Male | Female | Male |
| **Child age in months** | 25–36 | 3175 (29.8%) | 3343 (30.6%) | 3581 (30.3%) | 3934 (30.5%) | 4041 (33.5%) | 4456 (34.5%) | 22001 (33.4%) | 23430(33.3%) |
| | 13–24 | 3637 (34.1%) | 3730 (34.1%) | 3948 (33.4%) | 4222 (32.8%) | 3945 (32.7%) | 4302 (33.3%) | 21911 (33.2%) | 23273(33.0%) |
| | 7–12 | 1663 (15.6%) | 1715 (15.7%) | 1925 (16.3%) | 2118 (16.4%) | 2019 (16.8%) | 2150 (16.6%) | 10881 (16.5%) | 12007(17.1%) |
| | 0–6 | 2181 (20.5%) | 2153 (19.7%) | 2373 (20.1%) | 2611 (20.3%) | 2048(17%) | 2008 (15.5%) | 11114 (16.9%) | 11684(16.6%) |
| | Missing | 0(0%) | 0(0%) | 0(0%) | 0(0%) | 0(0%) | 0(0%) | 0(0%) | 0(0%) |
| **Type of caste** | SC | 1313 (12.3%) | 1445 (13.2%) | 2156 (18.2%) | 2382(18.5) | 2080 (17.3%) | 2292 (17.7%) | 12527(19%) | 13201(18.8%) |
| | ST | 1268 (11.9%) | 1262 (11.5%) | 1783 (15.1%) | 1822 (14.1%) | 2018 (16.7%) | 1935(15%) | 13292 (20.2%) | 13493(19.2%) |
| | OBC | - | - | 3419 (28.9%) | 3660 (28.4%) | 3930 (32.6%) | 4262(33%) | 25747 (39.1%) | 28179(40%) |
| | Other caste | 8075 (75.8%) | 8234 (75.3%) | 4387 (37.1%) | 4958 (38.5%) | 3513 (29.1%) | 3855 (29.8%) | 11362 (17.2%) | 12478(17.7%) |
| | Missing | 0(0%) | 0(0%) | 82(0.7%) | 63(0.5%) | 512(4.3%) | 572(4.4%) | 2979(4.5%) | 3043(4.3%) |
| **Mothers Education** | No education | 5927 (55.6%) | 6143 (56.1%) | 5784 (48.9%) | 6140 (47.7%) | 4731 (39.3%) | 4916 (38.1%) | 19260 (29.2%) | 20041(28.5%) |
| | <5 years | 739(6.9%) | 703(6.4%) | 1071(9.1%) | 1090(8.5%) | 867(7.2%) | 911(7.1%) | 4062(6.2%) | 4111(5.8%) |
| | 5–7 years | 1462 (13.7%) | 1409 (12.9%) | 1831 (15.5%) | 1856 (14.4%) | 1932(16%) | 1946 (15.1%) | 10321 (15.7%) | 11294(16%) |
| | 8–9 years | 938(8.8%) | 1068(9.8%) | 1210 (10.2%) | 1486 (11.5%) | 1812(15%) | 1965 (15.2%) | 12082 (18.3%) | 13279(18.9%) |
| | 10–11 years | 848(8.0%) | 862(7.9%) | 997(8.4%) | 1185(9.2%) | 1095(9.1%) | 1383 (10.7%) | 7638(11.6%) | 8075(11.5%) |
| | >12 years | 703(6.6%) | 71(6.6%) | 928(7.8%) | 1123(8.7%) | 1616 (13.4%) | 1795 (13.9%) | 12544(19%) | 13594(19.3%) |
| | Missing | 39(0.37%) | 39(0.4%) | 6(0.05%) | 5(0.04%) | 0(0%) | 0(0%) | 0(0%) | 0(0%) |
| **Size at birth** | (Small) | 2369 (22.2%) | 2129 (19.5%) | 3144 (26.6%) | 2970 (23.1%) | 2668 (22.1%) | 2511 (19.4%) | 8333(12.6%) | 7985(11.3%) |
| | Average | 6921 (64.9%) | 7177 (65.6%) | 7071 (59.8%) | 7858(61%) | 6612 (54.9%) | 7214 (55.9%) | 45296 (68.7%) | 48796(69.3%) |
| | Large | 1266 (11.9%) | 1564 (14.3%) | 1604 (13.6%) | 2051 (15.9%) | 2605 (21.6%) | 3021 (23.4%) | 10991 (16.7%) | 12349 (17.5%) |
| | Missing | 100(0.9%) | 71(0.6%) | 8(0.07%) | 6(0.05%) | 168(1.4%) | 170(1.3%) | 1287(2%) | 1264(1.8%) |
| **Birth order** | 4 or more | 3130 (29.4%) | 3270 (29.9%) | 3090(26.1) | 3395 (26.3%) | 2712 (22.5%) | 2893 (22.4%) | 10122 (15.4%) | 10800(15.3%) |
| | Three | 1918(18%) | 1951 (17.8%) | 2061 (17.4%) | 2286 (17.7%) | 1911 (15.9%) | 2147 (16.6%) | 10127 (15.4%) | 11487(16.3%) |
| | Second | 2669(25%) | 2668 (24.4%) | 3126 (26.4%) | 3467 (26.9%) | 3445 (28.6%) | 3762(29.1) | 20912 (31.7%) | 22162(31.5%) |
| | First | 2939 (27.6%) | 3052 (27.9%) | 3550(30%) | 3737(29%) | 3985 (33.1%) | 4114 (31.9%) | 24746 (37.5%) | 25945(36.9%) |
| | Missing | 0(0%) | 0(0%) | 0(0%) | 0(0%) | 0(0%) | 0(0%) | 0(0%) | 0(0%) |
| **Wealth Index** | Poorest | - | - | - | - | 2165(18%) | 2171 (16.8%) | 16886 (25.6%) | 17645(25.1%) |
| | Poorer | - | - | - | - | 2288(19%) | 2256 (17.5%) | 15704 (23.8%) | 16535(23.5%) |
| | Middle | - | - | - | - | 2487 (20.6%) | 2692 (20.8%) | 13318 (20.2%) | 14374(20.4%) |

*(Continued)*

**Table 1.** (Continued)

| Variable (n and %) in each category) | Category | NFHS-1 | | NFHS-2 | | NFHS-3 | | NFHS-4 | |
|---|---|---|---|---|---|---|---|---|---|
| | | Female | Male | Female | Male | Female | Male | Female | Male |
| | Richer | - | - | - | - | 2689 (22.3%) | 2948 (22.8%) | 11254 (17.1%) | 11886(16.9%) |
| | Richest | - | - | - | - | 2424 (20.1%) | 2849 (22.1%) | 8745(13.3%) | 9954(14.1%) |
| | Missing | - | - | - | - | 0(0%) | 0(0%) | 0(0%) | 0(0%) |
| **Mothers BMI** | <18.5 | - | - | 4152 (35.1%) | 4661 (36.2%) | 3970 (32.9%) | 4276(33%) | 16004 (24.3%) | 17596(25%) |
| | BMI>25 | - | - | 668(5.6%) | 699(5.4%) | 1044(8.7%) | 1208(9.4%) | 8377(12.7%) | 8972(12.7%) |
| | BMI 18.5–25 | - | - | 6957 (58.8%) | 7456 (57.9%) | 6959(57.7) | 7371(57%) | 41113 (62.4%) | 43421(61.7%) |
| | Missing | - | - | 50(0.4%) | 69(0.5%) | 80(0.7%) | 61(0.5%) | 413(0.6%) | 405(0.6%) |

-: No Data available, () percentage of data

From the quantile regression results, it was observed that child age in months for both girls and boys for the categories (25–38, 13–24, 7–12) was negatively associated and was statistically significant across the quantiles. If the child's age increases, the coefficient values decreases and these values are low in the lower quantiles compared to the upper quantiles. When the age and quantiles increases, the coefficient values decreases compared to the reference age group (0–6 months), it indicates that in the first six months of life, HAZ was better and when they grow older, the children become vulnerable to malnutrition. Although these values hold for all surveys, boys were found to have better HAZ values than the girls did across the quantiles; however, the coefficient values' strength is different in various quantiles.

The factor children's mother's education, while comparing the reference category (>12 years of education) with (No education, <5, 5–7, 8–9, 10–11 years) showed a negative association and statistically significant association in percentiles for the boys and girls, although girls whose mother having the education of 8–9 and 10-11-years education showed a positive association observed in Tables 3 and 4 for the 95th percentile. It was also observed that the children's HAZ coefficient values decreased when their mother's education increased from lower education to higher education. It indicates that, as the mothers get more educated, their children get less vulnerable to malnutrition.

A child's size at birth (for category: small and average) was found to be statistically significant and be negatively associated with HAZ scores across quantiles in the first, second, and fourth rounds of the survey. Suppose one moves from lower percentile to upper percentile, the value of HAZ increases. For the NFHS 3 child size at birth (category: average) was negatively associated with the reference category (Size at birth: Large). It indicates that the size at birth plays a main role in the area of child nutrition. Also observed that girls are having better HAZ outcomes when compared to boys. The size of a newborn baby at birth plays a significant role in malnutrition. This analysis showed that children are larger size at birth are less likely to be vulnerable to malnutrition as compared to the children who were smaller and of average size.

In all the rounds, the caste had a significant role. Children who belonged to the scheduled castes (SC) had a negative and statistically significant result compared to those belonging to other castes. In the third round, among the OBC children, males had a statistically negative significance (from 10th to 90th percentile) associated with the lower quantiles, and it was positively associated at 95th percentile. In fourth round, the type of caste (children who belonged

**Table 2. Quantile regression estimations for the different quantiles of the female (10517) and male (10830) children's, dependent variable (HAZ) NFHS-1.**

| Parameter | 0.05 Female | 0.05 Male | 0.1 Female | 0.1 Male | 0.25 Female | 0.25 Male | 0.5 Female | 0.5 Male | 0.75 Female | 0.75 Male | 0.9 Female | 0.9 Male | 0.95 Female | 0.95 Male |
|---|---|---|---|---|---|---|---|---|---|---|---|---|---|---|
| **Child age in Months** | | | | | | | | | | | | | | |
| 25-36in Months | -0.97* (-1.17,-0.78) | -0.63* (-0.83,-0.43) | -1.38* (-1.53,-1.23) | -0.94* (-1.14,-0.73) | -1.7* (-1.83,-1.57) | -1.28* (-1.40,-1.16) | -1.75* (-1.83,-1.65) | -1.56* (-1.67,-1.46) | -1.80* (-1.92,-1.66) | -1.8* (-1.92,-1.69) | -1.89* (-2.05,-1.72) | -1.93* (-2.12,-1.74) | -2.00* (-2.32,-1.70) | -1.80* (-2.06,-1.52) |
| 13-24 months | -0.68* (-0.89,-0.47) | -0.52* (-0.71,-0.32) | -1.01* (-1.17,-0.85) | -0.80* (-0.99,-0.60) | -1.32* (-1.44,-1.20) | -1.08* (-1.19,-0.97) | -1.40* (-1.49,-1.30) | -1.35* (-1.46,-1.26) | -1.40* (-1.51,-1.30) | -1.52* (-1.64,-1.40) | -1.55* (-1.72,-1.37) | -1.63* (-1.81,-1.44) | -1.57* (-1.90,-1.24) | -1.48* (-1.76,-1.2) |
| 7-12 months | -0.32* (-0.60,-0.04) | -0.25** (-0.50,-0.01) | -0.50* (-0.71,-0.28) | -0.33* (-0.55,-0.11) | -0.56* (-0.72,-0.40) | -0.40* (-0.53,-0.26) | -0.64* (-0.75,-0.54) | -0.50* (-0.64,-0.37) | -0.71* (-0.85,-0.56) | -0.67* (-0.81,-0.54) | -0.79* (-0.98,-0.58) | -0.78* (-1.02,-0.54) | -0.95* (-1.32,-0.57) | -0.67* (-0.95,-0.39) |
| 0-6 months | Reference | Reference | Reference | Reference | Reference | Reference | Reference | Reference | Reference | Reference | Reference | Reference | Reference | Reference |
| **Mothers Education** | | | | | | | | | | | | | | |
| No education | -0.68* (-1.10,-0.28) | -0.73* (-1.09,-0.37) | -1.03* (-1.31,-0.75) | -1.13* (-1.42,-0.83) | -1.21* (-1.41,-1.02) | -1.26* (-1.4,-1.09) | -1.05* (-1.19,-0.91) | -1.11* (-1.25,-0.98) | -0.76* (-0.94,-0.58) | -1.09* (-1.25,-0.94) | -0.74* (-0.97,-0.49) | -0.82* (-1.09,-0.55) | -0.46** (-0.82,-0.1) | -0.59* (-1.03,-0.15) |
| <5 years | -0.33 (-0.80,0.14) | -0.21 (-0.67,0.25) | -0.55* (-0.90,-0.20) | -0.49* (-0.87,-0.11) | -0.82* (-1.10,-0.57) | -0.77* (-0.96,-0.58) | -0.81* (-1.00,-0.61) | -0.80* (-0.97,-0.63) | -0.67* (-0.89,-0.45) | -0.77* (-0.98,-0.56) | -0.70* (-1.01,-0.37) | -0.70* (-1.08,-0.32) | -0.60** (-1.10,-0.10) | -0.48 (-1.13,0.17) |
| 5-7 years | -0.14 (-0.66,0.38) | -0.27*** (-0.71,0.16) | -0.35* (-0.67,-03) | -0.60* (-0.93,-0.26) | -0.59* (-0.80,-0.38) | -0.70* (-0.83,-0.51) | -0.61* (-0.74,-0.47) | -0.62* (-0.77,-0.46) | -0.44* (-0.63,-0.25) | -0.73* (-0.89,-0.57) | -0.60* (-0.86,-0.33) | -0.57* (-0.88,-0.25) | -0.43** (-0.87,0.01) | -0.39 (-0.92,0.14) |
| 8-9 years | -0.02 (-0.49,0.45) | -0.03 (-0.53,0.47) | -0.36* (-0.72,-.008) | -0.26*** (-0.62,0.10) | -0.45* (-0.67,-0.23) | -0.49* (-0.71,-0.27) | -0.47* (-0.63,-0.30) | -0.44* (-0.61,-0.28) | -0.29* (-0.49,-0.08) | -0.49* (-0.64,-0.33) | -0.55* (-0.83,-0.28) | -0.67* (-0.96,-0.38) | -0.68* (-1.08,-0.28) | -0.63* (-1.1,-0.16) |
| 10-11 years | -0.05 (-0.69,0.59) | 0.09 (-0.47,0.65) | -0.006 (-0.39,0.27) | -0.05 (-0.39,0.29) | -0.23** (-0.46,-0.01) | -0.25** (-0.45,-0.05) | -0.16*** (-0.32,0.01) | -0.25** (-0.43,-0.06) | -0.02 (-0.23,0.19) | -0.29* (-0.45,-0.14) | -0.21 (-0.57,0.14) | -0.33** (-0.68,0.02) | 0.25 (-0.29,0.79) | -0.17 (-0.66,0.33) |
| >12 years | Reference | Reference | Reference | Reference | Reference | Reference | Reference | Reference | Reference | Reference | Reference | Reference | Reference | Reference |
| **Size at birth** | | | | | | | | | | | | | | |
| Small | -0.35* (-0.59,-0.11) | -0.21*** (-0.43,0.014) | -0.34* (-0.51,-0.17) | -0.50* (-0.70,-0.31) | -0.45* (-0.60,-0.30) | -0.48* (-0.61,-0.35) | -0.53* (-0.65,-0.41) | -0.58* (-0.68,-0.46) | -0.57* (-0.74,-0.40) | -0.63* (-0.75,-0.50) | -0.69* (-0.90,-0.49) | -0.62* (-0.86,-0.38) | -0.94* (-1.30,-0.58) | -0.69* (-1.04,-0.34) |
| Average | -0.22*** (-0.46,-.02) | -0.04 (-0.23,0.16) | -0.24* (-0.39,-0.09) | -0.23* (-0.40,-0.06) | -0.23* (-0.37,-0.10) | -0.17* (-0.30,-0.04) | -0.29* (-0.39,-0.18) | -0.28* (-0.37,-0.18) | -0.32* (-0.46,-0.17) | -0.28* (-0.39,-0.17) | -0.39* (-0.57,-0.20) | -0.31* (-0.52,-0.1) | -0.39** (-0.72,-0.06) | -0.39** (-0.69,-0.08) |
| Large | Reference | Reference | Reference | Reference | Reference | Reference | Reference | Reference | Reference | Reference | Reference | Reference | Reference | Reference |
| Type of caste(SC) | -0.05 (-0.23,0.13) | -0.06 (-0.20,0.08) | -0.11 (-0.25,0.03) | -0.17** (-0.34,-0.001) | -0.15** (-0.28,-0.02) | -0.18* (-0.28,-0.08) | -0.13* (-0.23,-0.02) | -0.15* (-0.26,-0.04) | -0.23* (-0.37,-0.09) | -0.12*** (-0.24,0.12) | -0.18*** (-0.37,0.01) | -0.06 (-0.28,0.16) | -0.17 (-0.52,0.18) | -0.13 (-0.39,0.12) |
| ST | -0.33* (-0.57,-0.09) | -0.21*** (-0.43,0.0003) | -0.12 (-0.31,0.07) | -0.16*** (-0.34,0.019) | 0.12 (-0.03,0.27) | -0.07 (-0.20,0.06) | 0.17* (0.06,0.27) | 0.09*** (-0.02,0.22) | 0.24* (0.09,0.38) | 0.31* (0.12,0.48) | 0.53* (0.28,0.79) | 0.58* (0.25,0.9) | 0.93* (0.54,1.32) | 1.03* (0.64,1.41) |
| Other caste | Reference | Reference | Reference | Reference | Reference | Reference | Reference | Reference | Reference | Reference | Reference | Reference | Reference | Reference |
| **Birth order** | | | | | | | | | | | | | | |
| 4 or more | -0.13 (-0.31,0.05) | -0.19* (-0.36,-0.02) | -0.14*** (-0.29,.02) | -0.12 (-0.27,0.03) | -0.15** (-0.27,-0.03) | -0.08 (-0.18,0.024) | -0.13* (-0.22,-0.04) | -0.08 (-0.18,0.02) | -0.13** (-0.25,-0.01) | -0.04 (-0.16,0.08) | -0.002 (-0.17,0.17) | -0.11 (-0.33,0.11) | -0.09 (-0.40,0.21) | -0.15 (-0.52,0.21) |
| Three | -0.07 (-0.27,0.13) | -0.03 (-0.23,0.17) | -0.03 (-0.19,0.13) | -0.06 (-0.23,0.12) | 0.04 (-0.1,0.18) | 0.004 (-0.11,0.12) | -0.02 (-0.12,0.09) | -0.003 (-0.09,0.09) | -0.11*** (-0.24,0.016) | -0.08 (-0.2,0.04) | -0.11 (-0.29,0.07) | -0.1 (-0.32,0.14) | -0.33** (-0.62,-0.04) | -0.04 (-0.37,0.29) |
| Second | -0.03 (-0.22,0.16) | 0.02 (-0.17,0.21) | 0.01 (-0.13,0.15) | 0.04 (-0.12,0.20) | 0.05 (-0.08,0.18) | 0.04 (-0.056,0.14) | 0 | 0.013 (-0.073,0.1) | -0.02 (-0.14,0.09) | 0.04 (-0.07,0.15) | -0.01 (-0.17,0.14) | 0.06 (-0.14,0.26) | -0.11 (-0.41,0.19) | 0.1 (-0.2,0.41) |

*(Continued)*

**Table 2.** (Continued)

| Parameter | 0.05 | | 0.1 | | 0.25 | | 0.5 | | 0.75 | | 0.9 | | 0.95 | |
| | Female | Male | Female | Male | Female | Male | Female | Male | Female | Male | Female | Male | Female | Male |
|---|---|---|---|---|---|---|---|---|---|---|---|---|---|---|
| First | Reference | Reference | Reference | Reference | Reference | Reference | Reference | Reference | Reference | Reference | Reference | Reference | Reference | Reference |

*: 1% Significance

**: 5% Significance

***: 10% Significance; () includes 95% Confidence intervals

**Table 3. Quantile regression estimations for the different quantiles of the female (11682) and male (12742) children's, dependent variable (HAZ) NFHS-2.**

| Parameter | 0.05 | | 0.1 | | 0.25 | | 0.5 | | 0.75 | | 0.9 | | 0.95 | |
|---|---|---|---|---|---|---|---|---|---|---|---|---|---|---|
| | Female | Male | Female | Male | Female | Male | Female | Male | Female | Male | Female | Male | Female | Male |
| **Child age in Months** | | | | | | | | | | | | | | |
| 25–36in Months | -1.75* (-1.91,-1.58) | -1.32* (-1.5,-1.14) | -1.81* (-1.95,-1.67) | -1.48* (-1.61,-1.35) | -1.75* (-1.86,-1.65) | -1.51* (-1.62,-1.41) | -1.87* (-1.96,-1.78) | -1.63* (-1.72,-1.54) | -2.02* (-2.12,-1.91) | -1.89* (-1.99,-1.79) | -2.20* (-2.36,-2.03) | -2.24* (-2.4,-2.08) | -2.17* (-2.4,-1.94) | -2.28* (-2.49,-2.07) |
| 13–24 months | -1.52* (-1.67,-1.36) | -1.20* (-1.38,-1.01) | -1.45* (-1.59,-1.31) | -1.33* (-1.45,-1.21) | -1.41* (-1.51,-1.32) | -1.41* (-1.51,-1.32) | -1.51* (-1.6,-1.42) | -1.53* (-1.63,-1.44) | -1.72* (-1.83,-1.61) | -1.64* (-1.74,-1.54) | -1.87* (-2.03,-1.71) | -1.95* (-2.1,-1.81) | -1.93* (-2.18,-1.68) | -1.93* (-2.16,-1.7) |
| 7–12 months | -0.48* (-0.68,-0.27) | -0.54* (-0.76,-0.32) | -0.51* (-0.66,-0.36) | -0.59* (-0.75,-0.44) | -0.62* (-0.73,-0.51) | -0.61* (-0.73,-0.49) | -0.69* (-0.82,-0.58) | -0.67* (-0.78,-0.56) | -0.84* (-0.96,-0.72) | -0.78* (-0.9,-0.67) | -1.01* (-1.19,-0.83) | -1.03* (-1.23,-0.83) | -1.01* (-1.31,-0.71) | -0.96* (-1.21,-0.7) |
| 0–6 months | Reference | Reference | Reference | Reference | Reference | Reference | Reference | Reference | Reference | Reference | Reference | Reference | Reference | Reference |
| **Mothers Education** | | | | | | | | | | | | | | |
| No education | -1.22* (-1.47,-0.96) | -1.15* (-1.4,-0.89) | -1.21* (-1.4,-1.02) | -1.18* (-1.36,-0.99) | -1.11* (-1.25,-0.97) | -1.20* (-1.34,-1.07) | -1.04* (-1.16,-0.91) | -0.98* (-1.12,-0.85) | -0.82* (-0.96,-0.68) | -0.86* (-1,-0.73) | -0.66* (-0.84,-0.49) | -0.76* (-0.99,-0.53) | -0.38** (-0.7,-0.06) | -0.61* (-0.93,-0.29) |
| <5 years | -0.95* (-1.25,-0.65) | -0.79* (-1.13,-0.45) | -0.87* (-1.13,-0.61) | -0.74* (-0.93,-0.55) | -0.69* (-0.86,-0.51) | -0.81* (-0.96,-0.65) | -0.69* (-0.84,-0.53) | -0.65* (-0.81,-0.5) | -0.51* (-0.69,-0.32) | -0.64* (-0.8,-0.47) | -0.43* (-0.64,-0.21) | -0.77* (-1.05,-0.46) | -0.31 (-0.7,0.08) | -0.55* (-1.01,-0.09) |
| 5–7 years | -0.52* (-0.82,-0.22) | -0.73* (-1.01,-0.45) | -0.54* (-0.75,-0.33) | -0.81* (-1.03,-0.59) | -0.55* (-0.69,-0.42) | -0.71* (-0.87,-0.55) | -0.57* (-0.7,-0.44) | -0.67* (-0.83,-0.51) | -0.50* (-0.66,-0.33) | -0.55* (-0.71,-0.39) | -0.41* (-0.63,-0.2) | -0.60* (-0.84,-0.37) | -0.23 (-0.6,0.14) | -0.57* (-0.87,-0.28) |
| 8–9 years | -0.64* (-0.95,-0.32) | -0.52* (-0.84,-0.2) | -0.52* (-0.77,-0.27) | -0.49* (-0.71,-0.27) | -0.40* (-0.58,-0.23) | -0.49* (-0.64,-0.34) | -0.48* (-0.64,-0.32) | -0.37* (-0.52,-0.22) | -0.33* (-0.49,-0.17) | -0.33* (-0.49,-0.17) | -0.30** (-0.54,-0.07) | -0.42* (-0.67,-0.17) | -0.1 (-0.39,0.19) | -0.42** (-0.7,-0.15) |
| 10–11 years | -0.50* (-0.85,-0.16) | -0.46* (-0.81,-0.11) | -0.28** (-0.54,-0.02) | -0.35* (-0.57,-0.14) | -0.30* (-0.46,-0.14) | -0.35* (-0.49,-0.2) | -0.33* (-0.47,-0.19) | -0.25* (-0.41,-0.08) | -0.17*** (-0.34,0) | -0.2 (-0.36,-0.05) | -0.15 (-0.34,0.05) | -0.19 (-0.44,0.07) | -0.1 (-0.43,0.25) | -0.30*** (-0.62,0.02) |
| >12 years | Reference | Reference | Reference | Reference | Reference | Reference | Reference | Reference | Reference | Reference | Reference | Reference | Reference | Reference |
| **Size at birth** | | | | | | | | | | | | | | |
| Small | -0.42* (-0.6,-0.24) | -0.53* (-0.73,-0.33) | -0.45* (-0.59,-0.31) | -0.56* (-0.69,-0.42) | -0.55* (-0.66,-0.44) | -0.56* (-0.69,-0.42) | -0.60* (-0.72,-0.48) | -0.59* (-0.7,-0.49) | -0.52* (-0.65,-0.38) | -0.56* (-0.68,-0.45) | -0.50* (-0.7,-0.3) | -0.61* (-0.79,-0.42) | -0.49* (-0.73,-0.24) | -0.55* (-0.79,-0.3) |
| Average | -0.16*** (-0.34,0.02) | -0.27* (-0.44,-0.1) | -0.13*** (-0.28,0.02) | -0.33* (-0.45,-0.21) | -0.24* (-0.33,-0.14) | -0.34* (-0.43,-0.26) | -0.28* (-0.38,-0.18) | -0.32* (-0.42,-0.23) | -0.22* (-0.33,-0.1) | -0.32* (-0.41,-0.22) | -0.26* (-0.45,-0.07) | -0.42* (-0.59,-0.26) | -0.23*** (-0.45,-0.02) | -0.36* (-0.56,-0.15) |
| Large | Reference | Reference | Reference | Reference | Reference | Reference | Reference | Reference | Reference | Reference | Reference | Reference | Reference | Reference |
| **Type of caste** | | | | | | | | | | | | | | |
| (SC) | -0.21* (-0.39,-0.04) | -0.18** (-0.33,-0.03) | -0.16** (-0.29,-0.03) | -0.27* (-0.41,-0.13) | -0.19* (-0.3,-0.07) | -0.23* (-0.34,-0.13) | -0.19* (-0.29,-0.09) | -0.29* (-0.4,-0.18) | -0.18* (-0.28,-0.07) | -0.21* (-0.32,-0.11) | -0.20* (-0.37,-0.03) | -0.26* (-0.41,-0.11) | -0.27** (-0.53,-0.02) | -0.26** (-0.47,-0.06) |
| ST | -0.06 (-0.21,0.08) | -0.18*** (-0.34,-0.02) | 0.01 (-0.15,0.17) | -0.25* (-0.37,-0.12) | 0.12** (0.01,0.22) | -0.14** (-0.25,-0.03) | 0.056 (-0.05,0.16) | -0.13* (-0.23,-0.03) | 0.15** (0.02,0.28) | -0.04*** (-0.16,0.08) | 0.16** (-0.04,0.36) | 0.14*** (-0.08,0.36) | 0.30** (-0.06,0.65) | 0.43* (0.12,0.74) |
| OBC | -0.07 (-0.21,0.06) | -0.22* (-0.35,-0.09) | -0.14** (-0.24,-0.04) | -0.25* (-0.37,-0.13) | -0.05 (-0.14,0.03) | -0.11** (-0.19,-0.02) | -0.05 (-0.14,0.03) | -0.08*** (-0.16,0.01) | -0.09*** (-0.19,0.02) | -0.04 (-0.13,0.05) | -0.13*** (-0.25,0) | -0.07 (-0.2,0.05) | -0.20*** (-0.39,0) | -0.03 (-0.23,0.17) |
| Other caste | Reference | Reference | Reference | Reference | Reference | Reference | Reference | Reference | Reference | Reference | Reference | Reference | Reference | Reference |
| **Birth order** | | | | | | | | | | | | | | |
| 4 or more | -0.29* (-0.45,-0.13) | -0.22* (-0.37,-0.07) | -0.31* (-0.44,-0.19) | -0.25* (-0.37,-0.12) | -0.33* (-0.43,-0.23) | -0.23* (-0.34,-0.11) | -0.29* (-0.39,-0.2) | -0.23* (-0.33,-0.12) | -0.17* (-0.28,-0.05) | -0.13 (-0.23,-0.03) | 0.03 (-0.15,0.21) | 0.05 (-0.11,0.22) | 0.29** (0.01,0.57) | 0.17 (-0.09,0.43) |
| Three | -0.12 (-0.31,0.07) | -0.04 (-0.22,0.14) | -0.08 (-0.23,0.07) | -0.11*** (-0.27,0.04) | -0.13** (-0.23,-0.03) | -0.067 (-0.18,0.05) | -0.18* (-0.27,-0.1) | -0.09*** (-0.2,0.03) | -0.61 (-0.2,0.02) | -0.11*** (-0.21,0) | -0.02 (-0.19,0.15) | -0.14*** (-0.3,0.02) | -0.06 (-0.31,0.2) | -0.20*** (-0.41,0.02) |
| Second | -0.1 (-0.26,0.06) | -0.06 (-0.22,0.11) | -0.1 (-0.24,0.04) | -0.06 (-0.24,0.04) | -0.08*** (-0.17,0.01) | -0.11** (-0.19,-0.02) | -0.11** (-0.19,-0.03) | -0.06 (-0.14,0.02) | -0.07 (-0.16,0.02) | -0.05 (-0.15,0.05) | 0.03 (-0.1,0.17) | -0.03 (-0.17,0.12) | -0.03 (-0.23,0.17) | 0.08 (-0.11,0.27) |
| First | Reference | Reference | Reference | Reference | Reference | Reference | Reference | Reference | Reference | Reference | Reference | Reference | Reference | Reference |

(Continued)

**Table 3.** (Continued)

| Parameter | 0.05 Female | 0.05 Male | 0.1 Female | 0.1 Male | 0.25 Female | 0.25 Male | 0.5 Female | 0.5 Male | 0.75 Female | 0.75 Male | 0.9 Female | 0.9 Male | 0.95 Female | 0.95 Male |
|---|---|---|---|---|---|---|---|---|---|---|---|---|---|---|
| Mothers BMI | | | | | | | | | | | | | | |
| BMI<18.5 | -0.01 (-0.13,0.11) | -0.07 (-0.18,0.04) | -0.05 (-0.14,0.04) | -0.07 (-0.17,0.02) | -0.08** (-0.16,-0.01) | -0.09* (-0.17,-0.03) | -0.09* (-0.16,-0.03) | -0.17* (-0.24,-0.1) | -0.17* (-0.24,-0.09) | -0.24* (-0.31,-0.16) | -0.28* (-0.39,-0.17) | -0.28* (-0.39,-0.17) | -0.44* (-0.61,-0.28) | -0.50* (-0.65,-0.36) |
| BMI>25 | 0.40* (0.03,0.67) | 0.2 (-0.15,0.55) | 0.37* (0.09,0.65) | 0.27* (0.03,0.51) | 0.51* (0.33,0.68) | 0.29* (0.14,0.43) | 0.428* (0.29,0.57) | 0.24* (0.08,0.4) | 0.40* (0.2,0.59) | 0.28** (0.15,0.42) | 0.36* (0.14,0.57) | 0.28** (0.09,0.47) | 0.19 (-0.13,0.5) | 0.23 (-0.04,0.5) |
| BMI 18.5–25 | Reference | Reference | Reference | Reference | Reference | Reference | Reference | Reference | Reference | Reference | Reference | Reference | Reference | Reference |

*: 1% Significance

**: 5% Significance

***: 10% Significance; () includes 95% Confidence intervals

**Table 4. Quantile regression estimations for the different quantiles of the female (11315) and male (12133) children's, dependent variable (HAZ) NFHS-3.**

| Parameter | 0.05 Female | 0.05 Male | 0.10 Female | 0.10 Male | 0.25 Female | 0.25 Male | 0.50 Female | 0.50 Male | 0.75 Female | 0.75 Male | 0.90 Female | 0.90 Male | 0.95 Female | 0.95 Male |
|---|---|---|---|---|---|---|---|---|---|---|---|---|---|---|
| age in months | | | | | | | | | | | | | | |
| 25–36 Months | -0.86* (-1.1,-0.64) | -0.89* (-1.13,-0.66) | -1.05* (-1.19,-0.91) | -1.02* (-1.16,-0.88) | -1.36* (-1.46,-1.25) | -1.28* (-1.38,-1.19) | -1.51* (-1.6,-1.42) | -1.47* (-1.56,-1.39) | -1.68* (-1.78,-1.58) | -1.67* (-1.78,-1.57) | -1.67* (-1.85,-1.5) | -1.72* (-1.87,-1.56) | -1.88* (-2.13,-1.63) | -1.56* (-1.77,-1.34) |
| 13–24 months | -0.86* (-1.11,-0.61) | -0.87* (-1.07,-0.68) | -0.98* (-1.12,-0.84) | -1.01* (-1.15,-0.88) | -1.26* (-1.35,-1.16) | -1.28* (-1.39,-1.17) | -1.42* (-1.51,-1.33) | -1.45* (-1.55,-1.36) | -1.58* (-1.69,-1.48) | -1.58* (-1.68,-1.47) | -1.59* (-1.77,-1.41) | -1.59* (-1.77,-1.42) | -1.74* (-2,-1.48) | -1.41* (-1.63,-1.19) |
| 7–12 months | -0.16 (-0.42,0.1) | -0.26** (-0.48,-0.03) | -0.35* (-0.53,-0.18) | -0.41* (-0.6,-0.22) | -0.46* (-0.58,-0.33) | -0.56* (-0.69,-0.42) | -0.54* (-0.66,-0.42) | -0.63* (-0.74,-0.52) | -0.57* (-0.7,-0.45) | -0.68* (-0.79,-0.57) | -0.56* (-0.78,-0.35) | -0.70* (-0.87,-0.52) | -0.61* (-0.96,-0.27) | -0.46* (-0.74,-0.18) |
| 0–6 months | Reference | Reference | Reference | Reference | Reference | Reference | Reference | Reference | Reference | Reference | Reference | Reference | Reference | Reference |
| Mothers Education | | | | | | | | | | | | | | |
| No education | -0.79* (-1.06,-0.51) | -0.50* (-0.8,-0.19) | -0.64* (-0.84,-0.43) | -0.55* (-0.74,-0.34) | -0.51* (-0.65,-0.37) | -0.58* (-0.72,-0.44) | -0.55* (-0.68,-0.42) | -0.63* (-0.75,-0.49) | -0.42* (-0.59,-0.26) | -0.51* (-0.65,-0.37) | -0.26** (-0.51,-0.01) | -0.52* (-0.75,-0.3) | -0.04 (-0.42,0.35) | -0.57* (-0.87,-0.27) |
| <5 years | -0.90* (-1.26,-0.53) | -0.19 (-0.57,0.2) | -0.59* (-0.83,-0.35) | -0.14 (-0.38,0.11) | -0.47* (-0.62,-0.31) | -0.28* (-0.5,-0.08) | -0.58* (-0.72,-0.42) | -0.37* (-0.53,-0.21) | -0.42* (-0.62,-0.22) | -0.31* (-0.5,-0.13) | -0.2 (-0.62,0.22) | -0.26*** (-0.55,0.03) | -0.03 (-0.52,0.46) | -0.12 (-0.53,0.29) |
| 5–7 years | -0.31** (-0.61,-0.01) | -0.30** (-0.6,0.01) | -0.24* (-0.41,-0.06) | -0.36* (-0.56,-0.15) | -0.24* (-0.38,-0.11) | -0.40* (-0.54,-0.27) | -0.32* (-0.45,-0.19) | -0.50* (-0.63,-0.37) | -0.30* (-0.45,-0.13) | -0.41* (-0.55,-0.26) | -0.29** (-0.55,-0.04) | -0.42* (-0.64,-0.19) | -0.16 (-0.47,0.16) | -0.39** (-0.7,-0.07) |
| 8–9 years | -0.31** (-0.61,-0.01) | -0.15 (-0.45,0.15) | -0.21** (-0.39,-0.02) | -0.14 (-0.33,0.07) | -0.18** (-0.31,-0.04) | -0.24* (-0.37,-0.1) | -0.24* (-0.36,-0.11) | -0.39* (-0.52,-0.27) | -0.21* (-0.36,-0.06) | -0.34* (-0.48,-0.19) | -0.19 (-0.42,0.04) | -0.33* (-0.56,-0.1) | 0.06 (-0.32,0.44) | -0.11 (-0.46,0.24) |
| 10–11 years | -0.12 (-0.45,0.22) | -0.21*** (-0.56,0.15) | 0.05 (-0.13,0.21) | -0.13 (-0.35,0.1) | -0.07 (-0.22,0.07) | -0.15*** (-0.28,-0.01) | -0.17* (-0.3,-0.05) | -0.33* (-0.45,-0.21) | -0.22* (-0.37,-0.07) | -0.27* (-0.4,-0.12) | -0.21 (-0.48,0.05) | -0.29* (-0.48,-0.09) | 0.07 (-0.26,0.39) | -0.33* (-0.64,-0.02) |
| >12 years | Reference | Reference | Reference | Reference | Reference | Reference | Reference | Reference | Reference | Reference | Reference | Reference | Reference | Reference |
| Size at birth | | | | | | | | | | | | | | |
| Small | -0.32* (-0.53,-0.12) | -0.31* (-0.5,-0.13) | -0.35* (-0.46,-0.23) | -0.39* (-0.54,-0.23) | -0.35* (-0.44,-0.26) | -0.40* (-0.51,-0.28) | -0.27* (-0.35,-0.18) | -0.45* (-0.55,-0.36) | -0.19* (-0.3,-0.07) | -0.44* (-0.55,-0.33) | -0.19*** (-0.36,-0.01) | -0.42* (-0.59,-0.26) | -0.13 (-0.38,0.11) | -0.49* (-0.74,-0.24) |
| Average | -0.11 (-0.27,0.06) | -0.03 (-0.19,0.14) | -0.10*** (-0.19,0.01) | -0.03 (-0.15,0.1) | -0.12* (-0.2,-0.04) | -0.10** (-0.19,-0.02) | -0.06 (-0.14,0.01) | -0.15* (-0.22,-0.07) | -0.03 (-0.12,0.06) | -0.19* (-0.27,-0.1) | -0.08 (-0.22,0.06) | -0.16** (-0.29,-0.03) | 0.01 (-0.21,0.23) | -0.18*** (-0.38,0.03) |
| Large | Reference | Reference | Reference | Reference | Reference | Reference | Reference | Reference | Reference | Reference | Reference | Reference | Reference | Reference |
| Type of caste | | | | | | | | | | | | | | |
| (SC) | -0.25* (-0.45,-0.06) | -0.51* (-0.69,-0.32) | -0.16* (-0.3,-0.02) | -0.23* (-0.38,-0.08) | -0.24* (-0.34,-0.15) | -0.24* (-0.34,-0.13) | -0.24* (-0.33,-0.14) | -0.23* (-0.32,-0.14) | -0.25* (-0.35,-0.14) | -0.23* (-0.32,-0.13) | -0.09 (-0.27,0.09) | -0.19* (-0.38,0) | 0.04 (-0.24,0.31) | 0.01 (-0.25,0.25) |
| ST | 0.04 (-0.19,0.27) | -0.04 (-0.27,0.19) | 0.01 (-0.13,0.15) | -0.09 (-0.23,0.05) | 0.05 (-0.07,0.15) | -0.06 (-0.16,0.07) | 0.09*** (0,0.17) | -0.04 (-0.13,0.05) | 0.16* (0.04,0.28) | -0.07 (-0.16,0.04) | 0.34* (0.12,0.55) | 0.14 (-0.09,0.36) | 0.54* (0.23,0.85) | 0.41* (0.11,0.72) |
| OBC | -0.14*** (-0.33,0.05) | -0.1 (-0.26,0.06) | -0.14* (-0.26,-0.02) | -0.17* (-0.3,-0.05) | -0.13* (-0.21,-0.05) | -0.15* (-0.24,-0.05) | -0.12* (-0.2,-0.04) | -0.15* (-0.22,-0.07) | -0.12** (-0.21,-0.03) | -0.14* (-0.22,-0.05) | 0.01 (-0.14,0.15) | -0.05 (-0.19,0.08) | 0.05 (-0.15,0.26) | 0.12 (-0.08,0.32) |
| Other caste | Reference | Reference | Reference | Reference | Reference | Reference | Reference | Reference | Reference | Reference | Reference | Reference | Reference | Reference |
| Birth order | | | | | | | | | | | | | | |
| (4 or more) | -0.25* (-0.45,-0.06) | -0.51* (-0.69,-0.32) | -0.29* (-0.46,-0.13) | -0.37* (-0.52,-0.22) | -0.25* (-0.36,-0.15) | -0.25* (-0.36,-0.14) | -0.18* (-0.27,-0.09) | -0.18* (-0.27,-0.09) | -0.16* (-0.28,-0.05) | -0.08 (-0.19,0.04) | -0.07 (-0.25,0.12) | 0.14*** (-0.07,0.34) | 0.09 (-0.19,0.37) | 0.34* (0.07,0.61) |
| Three | -0.15 (-0.36,0.05) | -0.24* (-0.43,-0.04) | -0.18* (-0.33,-0.03) | -0.17* (-0.32,-0.03) | -0.11** (-0.21,-0.01) | -0.16* (-0.27,-0.04) | -0.05 (-0.15,0.04) | -0.16* (-0.23,-0.06) | -0.03 (-0.14,0.1) | -0.12** (-0.22,-0.01) | 0.04 (-0.2,0.28) | -0.05 (-0.24,0.15) | 0.05 (-0.28,0.37) | 0.23*** (-0.06,0.54) |
| Second | -0.04 (-0.2,0.13) | -0.22* (-0.39,-0.04) | -0.08 (-0.17,0.02) | -0.07 (-0.21,0.06) | -0.11** (-0.18,-0.03) | -0.05 (-0.14,0.04) | -0.04 (-0.12,0.03) | -0.05 (-0.14,0.04) | -0.03 (-0.12,0.06) | -0.01 (-0.11,0.09) | 0.02 (-0.13,0.17) | 0.04 (-0.11,0.19) | 0.01 (-0.22,0.24) | -0.03 (-0.22,0.17) |
| First | Reference | Reference | Reference | Reference | Reference | Reference | Reference | Reference | Reference | Reference | Reference | Reference | Reference | Reference |
| Mothers BMI | | | | | | | | | | | | | | |

(Continued)

**Table 4.** (Continued)

| Parameter | 0.05 Female | 0.05 Male | 0.10 Female | 0.10 Male | 0.25 Female | 0.25 Male | 0.50 Female | 0.50 Male | 0.75 Female | 0.75 Male | 0.90 Female | 0.90 Male | 0.95 Female | 0.95 Male |
|---|---|---|---|---|---|---|---|---|---|---|---|---|---|---|
| BMI<18.5 | -0.07 (-0.22,0.08) | -0.14** (-0.28,0.03) | -0.08*** (-0.18,0.02) | -0.14** (-0.25,-0.03) | -0.16* (-0.23,-0.09) | -0.17* (-0.25,-0.08) | -0.21* (-0.27,-0.15) | -0.18* (-0.25,-0.11) | -0.18* (-0.27,-0.1) | -0.18* (-0.26,-0.1) | -0.29* (-0.44,-0.18) | -0.18 (-0.32,-0.04) | -0.41* (-0.61,-0.22) | -0.24* (-0.42,-0.05) |
| BMI>25 | 0.11 (-0.21,0.43) | 0.08 (-0.2,0.35) | 0.16** (-0.01,0.32) | 0.07 (-0.12,0.25) | 0.18* (0.04,0.3) | 0.09 (-0.02,0.19) | 0.16* (0.05,0.27) | 0.12** (0.01,0.23) | 0.25* (0.09,0.41) | 0.11 (-0.01,0.22) | 0.27** (0.06,0.5) | 0.01 (-0.14,0.17) | 0.01 (-0.26,0.29) | -0.07 (-0.35,0.22) |
| BMI 18.5–25 | Reference | Reference | Reference | Reference | Reference | Reference | Reference | Reference | Reference | Reference | Reference | Reference | Reference | Reference |
| Wealth Index | | | | | | | | | | | | | | |
| (Poorest) | -0.73* (-0.99,-0.48) | -0.72* (-1.01,-0.44) | -0.79* (-0.96,-0.62) | -0.82* (-1.05,-0.61) | -0.73* (-0.86,-0.6) | -0.80* (-0.95,-0.65) | -0.64* (-0.78,-0.5) | -0.66* (-0.81,-0.5) | -0.64* (-0.83,-0.45) | -0.61* (-0.78,-0.45) | -0.54* (-0.86,-0.23) | -0.53* (-0.8,-0.26) | -0.67* (-1.09,-0.26) | -0.46* (-0.86,-0.06) |
| Poorer | -0.52* (-0.8,-0.23) | -0.73* (-0.97,-0.49) | -0.52* (-0.68,-0.33) | -0.71* (-0.93,-0.49) | -0.52* (-0.64,-0.4) | -0.60* (-0.74,-0.45) | -0.48* (-0.6,-0.36) | -0.57* (-0.7,-0.44) | -0.46* (-0.63,-0.29) | -0.50* (-0.63,-0.36) | -0.44* (-0.69,-0.2) | -0.47* (-0.7,-0.23) | -0.47* (-0.86,-0.07) | -0.43* (-0.81,-0.06) |
| Middle | -0.40* (-0.66,-0.15) | -0.50* (-0.76,-0.25) | -0.49* (-0.64,-0.34) | -0.53* (-0.73,-0.34) | -0.42* (-0.53,-0.31) | -0.48* (-0.61,-0.36) | -0.39* (-0.5,-0.29) | -0.31* (-0.43,-0.2) | -0.38* (-0.51,-0.23) | -0.37* (-0.5,-0.25) | -0.31* (-0.55,-0.07) | -0.44* (-0.64,-0.23) | -0.43** (-0.79,-0.07) | -0.52* (-0.84,-0.21) |
| Richer | -0.46* (-0.69,-0.23) | -0.26*** (-0.49,-0.03) | -0.40* (-0.53,-0.26) | -0.30* (-0.45,-0.13) | -0.26* (-0.37,-0.16) | -0.30* (-0.42,-0.18) | -0.24* (-0.35,-0.13) | -0.20* (-0.3,-0.09) | -0.25* (-0.39,-0.11) | -0.22* (-0.35,-0.1) | -0.16 (-0.37,0.06) | -0.22* (-0.38,-0.05) | -0.38* (-0.65,-0.11) | -0.40* (-0.63,-0.17) |
| Richest | Reference | Reference | Reference | Reference | Reference | Reference | Reference | Reference | Reference | Reference | Reference | Reference | Reference | Reference |

*: 1% Significance

**: 5% Significance

***: 10% Significance; () includes 95% Confidence intervals

**Table 5. Quantile regression estimations for the different quantiles of the female (61582) and male (66018) children's, dependent variable (HAZ). NFHS-4.**

| Parameter | 0.05 Female | 0.05 Male | 0.1 Female | 0.1 Male | 0.25 Female | 0.25 Male | 0.5 Female | 0.5 Male | 0.75 Female | 0.75 Male | 0.9 Female | 0.9 Male | 0.95 Female | 0.95 Male |
|---|---|---|---|---|---|---|---|---|---|---|---|---|---|---|
| **Child age in Months** | | | | | | | | | | | | | | |
| 25–36 Months | -0.50* (-0.59,-0.4) | -0.12* (-0.2,-0.02) | -0.67* (-0.73,-0.6) | -0.36* (-0.43,-0.28) | -1.02* (-1.07,-0.97) | -0.82* (-0.86,-0.77) | -1.23* (-1.27,-1.19) | -1.15* (-1.19,-1.12) | -1.41* (-1.46,-1.35) | -1.36* (-1.41,-1.31) | -1.42* (-1.5,-1.34) | -1.41* (-1.5,-1.33) | -1.37* (-1.5,-1.21) | -1.37* (-1.51,-1.24) |
| 13–24 months | -0.52* (-0.61,-0.44) | -0.41* (-0.5,-0.31) | -0.68* (-0.74,-0.61) | -0.59* (-0.67,-0.51) | -1.00* (-1.05,-0.95) | -1.00* (-1.04,-0.95) | -1.15* (-1.19,-1.11) | -1.21* (-1.26,-1.17) | -1.27* (-1.32,-1.21) | -1.28* (-1.33,-1.24) | -1.20* (-1.28,-1.12) | -1.26* (-1.33,-1.18) | -1.19* (-1.33,-1.04) | -1.15* (-1.27,-1.04) |
| 7–12 months | -0.07 (-0.19,0.03) | -0.11** (-0.22,0.01) | -0.13* (-0.21,-0.05) | -0.11* (-0.2,-0.01) | -0.28* (-0.33,-0.23) | -0.29* (-0.35,-0.23) | -0.35* (-0.4,-0.3) | -0.35* (-0.4,-0.3) | -0.38* (-0.45,-0.31) | -0.37* (-0.43,-0.3) | -0.32* (-0.43,-0.21) | -0.30* (-0.39,-0.21) | -0.27* (-0.45,-0.08) | -0.28* (-0.43,-0.14) |
| 0–6 months | Reference | Reference | Reference | Reference | Reference | Reference | Reference | Reference | Reference | Reference | Reference | Reference | Reference | Reference |
| **Mothers Education** | | | | | | | | | | | | | | |
| No education | -0.50* (-0.6,-0.38) | -0.46* (-0.56,-0.35) | -0.53* (-0.61,-0.45) | -0.46* (-0.54,-0.38) | -0.48* (-0.53,-0.43) | -0.47* (-0.52,-0.41) | -0.43* (-0.48,-0.38) | -0.46* (-0.51,-0.41) | -0.34* (-0.41,-0.28) | -0.42* (-0.48,-0.35) | -0.27* (-0.37,-0.17) | -0.31* (-0.41,-0.2) | -0.15 (-0.36,0.04) | -0.31* (-0.48,-0.15) |
| <5 years | -0.34* (-0.49,-0.2) | -0.27* (-0.4,-0.13) | -0.36* (-0.47,-0.25) | -0.26* (-0.35,-0.14) | -0.37* (-0.45,-0.3) | -0.31* (-0.37,-0.23) | -0.32* (-0.4,-0.24) | -0.35* (-0.42,-0.28) | -0.32* (-0.41,-0.24) | -0.38* (-0.46,-0.29) | -0.26* (-0.41,-0.11) | -0.40* (-0.54,-0.25) | -0.30** (-0.56,-0.03) | -0.38* (-0.66,-0.13) |
| 5–7 years | -0.30* (-0.41,-0.2) | -0.24* (-0.35,-0.14) | -0.30* (-0.38,-0.21) | -0.29* (-0.37,-0.22) | -0.30* (-0.35,-0.25) | -0.32* (-0.37,-0.27) | -0.31* (-0.36,-0.27) | -0.31* (-0.36,-0.27) | -0.29* (-0.36,-0.24) | -0.33* (-0.39,-0.28) | -0.24* (-0.36,-0.13) | -0.30* (-0.41,-0.2) | -0.17*** (-0.41,0.03) | -0.29* (-0.46,-0.13) |
| 8–9 years | -0.09** (-0.19,0.02) | -0.07 (-0.19,0.04) | -0.17* (-0.24,-0.08) | -0.12* (-0.19,-0.05) | -0.21* (-0.25,-0.15) | -0.17* (-0.22,-0.12) | -0.25* (-0.29,-0.21) | -0.22* (-0.27,-0.18) | -0.24* (-0.3,-0.19) | -0.24* (-0.29,-0.19) | -0.24* (-0.34,-0.14) | -0.27* (-0.35,-0.18) | -0.20** (-0.37,-0.04) | -0.30* (-0.46,-0.15) |
| 10–11 years | -0.01 (-0.13,0.1) | -0.08 (-0.24,0.08) | -0.10** (-0.17,-0.01) | -0.08*** (-0.15,0) | -0.11* (-0.15,-0.05) | -0.09* (-0.14,-0.03) | -0.11* (-0.16,-0.06) | -0.12* (-0.17,-0.08) | -0.12* (-0.19,-0.05) | -0.10* (-0.16,-0.05) | -0.10*** (-0.21,0.01) | -0.07 (-0.17,0.02) | -0.08 (-0.3,0.09) | -0.1 (-0.28,0.07) |
| >12 years | Reference | Reference | Reference | Reference | Reference | Reference | Reference | Reference | Reference | Reference | Reference | Reference | Reference | Reference |
| **Size at birth** | | | | | | | | | | | | | | |
| Small | -0.35* (-0.46,-0.25) | -0.22* (-0.32,-0.11) | -0.36* (-0.45,-0.28) | -0.27* (-0.34,-0.19) | -0.36* (-0.42,-0.31) | -0.41* (-0.45,-0.35) | -0.41* (-0.46,-0.36) | -0.45* (-0.5,-0.41) | -0.48* (-0.54,-0.42) | -0.47* (-0.53,-0.4) | -0.61* (-0.71,-0.5) | -0.54* (-0.63,-0.45) | -0.64* (-0.86,-0.45) | -0.62* (-0.76,-0.48) |
| Average | 0.02 (-0.06,0.1) | -0.02 (-0.09,0.05) | -0.002 (-0.06,0.05) | -0.04 (-0.1,0.02) | -0.05** (-0.09,0.01) | -0.11* (-0.14,-0.07) | -0.08* (-0.12,-0.04) | -0.13* (-0.16,-0.1) | -0.13* (-0.18,-0.09) | -0.17* (-0.21,-0.12) | -0.2* (-0.28,-0.12) | -0.24* (-0.31,-0.17) | -0.19** (-0.34,-0.05) | -0.29* (-0.41,-0.18) |
| Large | Reference | Reference | Reference | Reference | Reference | Reference | Reference | Reference | Reference | Reference | Reference | Reference | Reference | Reference |
| **Type of caste** | | | | | | | | | | | | | | |

*(Continued)*

**Table 5.** (Continued)

| Parameter | 0.05 Female | 0.05 Male | 0.1 Female | 0.1 Male | 0.25 Female | 0.25 Male | 0.5 Female | 0.5 Male | 0.75 Female | 0.75 Male | 0.9 Female | 0.9 Male | 0.95 Female | 0.95 Male |
|---|---|---|---|---|---|---|---|---|---|---|---|---|---|---|
| (SC) | -0.10** (-0.2,-0.01) | -0.19* (-0.28,-0.1) | -0.17* (-0.25,-0.1) | -0.24* (-0.31,-0.17) | -0.20* (-0.24,-0.15) | -0.23* (-0.28,-0.18) | -0.24* (-0.28,-0.2) | -0.22* (-0.27,-0.18) | -0.24* (-0.29,-0.19) | -0.22* (-0.28,-0.17) | -0.21* (-0.3,-0.11) | -0.19* (-0.29,-0.1) | -0.31* (-0.46,-0.13) | -0.13*** (-0.27,0.01) |
| ST | 0.01 (-0.09,0.11) | -0.20* (-0.3,-0.11) | -0.01 (-0.09,0.07) | -0.18* (-0.25,-0.1) | 0.001 (-0.05,0.05) | -0.09* (-0.14,-0.04) | 0.02 (-0.03,0.07) | -0.07* (-0.12,-0.02) | 0.05*** (-0.01,0.11) | -0.07* (-0.13,-0.02) | 0.14* (0.04,0.25) | 0.05 (-0.05,0.15) | 0.29* (0.1,0.48) | 0.18** (0.02,0.33) |
| OBC | -0.06 (-0.14,0.02) | -0.12* (-0.21,-0.05) | -0.12* (-0.18,-0.05) | -0.14* (-0.2,-0.08) | -0.15* (-0.19,-0.1) | -0.16* (-0.2,-0.12) | -0.16* (-0.2,-0.12) | -0.16* (-0.2,-0.13) | -0.18* (-0.23,-0.13) | -0.20* (-0.24,-0.16) | -0.16* (-0.24,-0.08) | -0.22* (-0.3,-0.15) | -0.17** (-0.29,-0.03) | -0.23* (-0.35,-0.11) |
| Other caste | Reference | Reference | Reference | Reference | Reference | Reference | Reference | Reference | Reference | Reference | Reference | Reference | Reference | Reference |
| Birth order | | | | | | | | | | | | | | |
| (4 or more) | -0.30* (-0.39,-0.19) | -0.12** (-0.21,-0.03) | -0.25* (-0.32,-0.17) | -0.14* (-0.21,-0.07) | -0.23* (-0.27,-0.18) | -0.16* (-0.21,-0.11) | -0.21* (-0.25,-0.15) | -0.12* (-0.17,-0.08) | -0.18* (-0.24,-0.13) | -0.11* (-0.17,-0.05) | -0.13* (-0.24,0.03) | -0.02 (-0.11,0.07) | -0.1 (-0.29,0.09) | 0.07 (-0.07,0.23) |
| Three | -0.17* (-0.28,0.06) | -0.10** (-0.19,-0.01) | -0.18* (-0.24,-0.11) | -0.11* (-0.18,-0.04) | -0.15* (-0.19,-0.1) | -0.10* (-0.14,-0.05) | -0.14* (-0.17,-0.09) | -0.09* (-0.13,-0.05) | -0.11* (-0.17,-0.06) | -0.07* (-0.12,-0.03) | -0.06 (-0.15,0.03) | -0.04 (-0.12,0.05) | 0.03 (-0.14,0.2) | 0.02 (-0.1,0.16) |
| Second | -0.08** (-0.16,0) | 0.02 (-0.05,0.09) | -0.10* (-0.15,-0.03) | -0.03 (-0.09,0.02) | -0.07* (-0.11,-0.04) | -0.06* (-0.1,-0.02) | -0.07* (-0.1,-0.04) | -0.04* (-0.08,-0.01) | -0.05* (-0.1,-0.01) | -0.03 (-0.07,0.01) | -0.08* (-0.15,0) | 0.03 (-0.05,0.09) | -0.01 (-0.13,0.13) | 0.10*** (-0.01,0.2) |
| First | Reference | Reference | Reference | Reference | Reference | Reference | Reference | Reference | Reference | Reference | Reference | Reference | Reference | Reference |
| Mothers BMI | | | | | | | | | | | | | | |
| BMI<18.5 | -0.11* (-0.18,-0.03) | - | -0.13* (-0.18,-0.07) | -0.10* (-0.15,-0.05) | -0.16* (-0.19,-0.12) | -0.15* (-0.19,-0.12) | -0.20* (-0.23,-0.17) | -0.19* (-0.22,-0.16) | -0.27* (-0.3,-0.22) | -0.21* (-0.25,-0.18) | -0.42* (-0.48,-0.35) | -0.34* (-0.4,-0.29) | -0.57* (-0.67,-0.44) | -0.46* (-0.56,-0.38) |
| BMI>25 | 0.28* (0.18,0.37) | 0.26* (0.17,0.35) | 0.24* (0.16,0.3) | 0.25* (0.18,0.32) | 0.20* (0.16,0.24) | 0.19* (0.14,0.23) | 0.15* (0.11,0.19) | 0.13* (0.08,0.17) | 0.09* (0.04,0.15) | 0.12* (0.07,0.17) | 0.02 (-0.07,0.12) | 0.08*** (-0.01,0.16) | -0.11 (-0.26,0.03) | 0.03 (-0.09,0.16) |
| BMI 18.5-25 | Reference | Reference | Reference | Reference | Reference | Reference | Reference | Reference | Reference | Reference | Reference | Reference | Reference | Reference |
| Wealth Index | | | | | | | | | | | | | | |
| (Poorest) | -0.62* (-0.75,-0.49) | -0.56* (-0.69,-0.44) | -0.67* (-0.77,-0.59) | -0.62* (-0.72,-0.52) | -0.68* (-0.74,-0.62) | -0.63* (-0.69,-0.57) | -0.60* (-0.66,-0.54) | -0.58* (-0.63,-0.52) | -0.57* (-0.64,-0.5) | -0.51* (-0.58,-0.45) | -0.54* (-0.66,-0.43) | -0.47* (-0.58,-0.37) | -0.53* (-0.76,-0.3) | -0.46* (-0.63,-0.28) |
| Poorer | -0.48* (-0.6,-0.36) | -0.31* (-0.43,-0.2) | -0.47* (-0.57,-0.39) | -0.40* (-0.48,-0.32) | -0.46* (-0.51,-0.41) | -0.39* (-0.45,-0.34) | -0.45* (-0.5,-0.4) | -0.38* (-0.44,-0.34) | -0.46* (-0.53,-0.39) | -0.39* (-0.45,-0.33) | -0.47* (-0.58,-0.37) | -0.42* (-0.52,-0.32) | -0.46* (-0.67,-0.26) | -0.39* (-0.55,-0.2) |
| Middle | -0.31* (-0.41,-0.2) | -0.20* (-0.33,-0.07) | -0.33* (-0.42,-0.25) | -0.17* (-0.25,-0.08) | -0.32* (-0.37,-0.27) | -0.23* (-0.28,-0.18) | -0.32* (-0.36,-0.28) | -0.25* (-0.29,-0.2) | -0.34* (-0.4,-0.27) | -0.24* (-0.31,-0.19) | -0.38* (-0.48,-0.28) | -0.25* (-0.35,-0.15) | -0.34* (-0.55,-0.11) | -0.18** (-0.35,0) |

(Continued)

**Table 5.** (Continued)

| Parameter | 0.05 | | 0.1 | | 0.25 | | 0.5 | | 0.75 | | 0.9 | | 0.95 | |
|---|---|---|---|---|---|---|---|---|---|---|---|---|---|---|
| | Female | Male | Female | Male | Female | Male | Female | Male | Female | Male | Female | Male | Female | Male |
| Richer | -0.20* (-0.31,-0.09) | -0.08 (-0.19,0.03) | -0.18* (-0.26,-0.1) | -0.09** (-0.17,-0.01) | -0.19* (-0.24,-0.14) | -0.10* (-0.15,-0.05) | -0.17* (-0.22,-0.12) | -0.11* (-0.16,-0.06) | -0.20* (-0.26,-0.14) | -0.12* (-0.17,-0.06) | -0.30* (-0.41,-0.2) | -0.11** (-0.2,-0.02) | -0.23** (-0.43,-0.02) | -0.07 (-0.22,0.09) |
| Richest | Reference | Reference | Reference | Reference | Reference | Reference | Reference | Reference | Reference | Reference | Reference | Reference | Reference | Reference |

*: 1% Significance

**: 5% Significance

***: 10% Significance; () includes 95% Confidence intervals

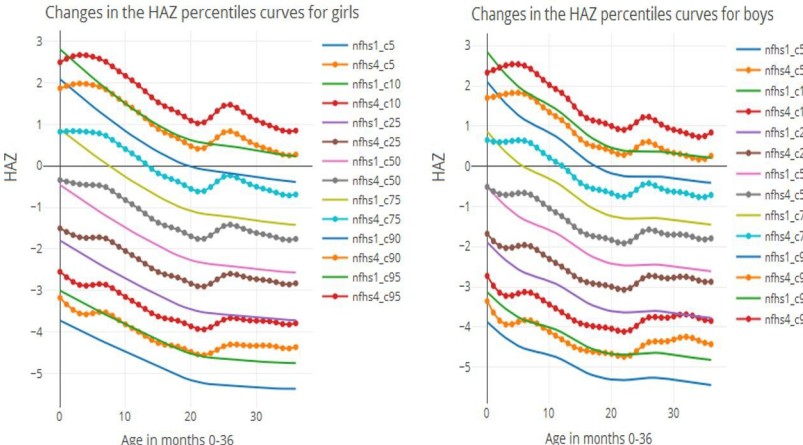

**Fig 1. LMS curves for child aged 0–36 months in NFHS 1 (1992) and NFHS 4 (2016).**

to OBC) in male children had a negative statistically significant association (except 10th percentile).

In all, the rounds of NFHS survey Birth order (category: four or more, 3) was found to have a negative association and statistically significant in the lower tails than the upper tails of distribution compared with the reference category. It also revealed that children were more prone to be nutritionally vulnerable when the birth order increased from two to four. Mothers, nutritional status (category: BMI<18.5) showed negatively significant association in the second, third and fourth round of NFHS and these coefficient values are low in the lower tails compared to the upper tails of the distribution. It indicates that children whose mothers BMI <18.5 are more likely to be stunted than their best peers are. Children whose mothers BMI (>25) had a positive association in all percentiles compared to the reference category. It reveals that the higher BMI values of mother's children are less likely to stunted when compared with children's mother's nutritional status is normal, i.e., BMI (18.5–25). In this study, the wealth index was calculated for the three and fourth rounds of the survey due to no data available for the first and second rounds of the survey. The wealth index for both genders for the categories (categories: poorest, poorer, middle, richer) was negatively and statistically significant with the whole percentile of the distribution. If the child belonging to the wealth status was the poorest, the coefficient values were high in lower tails and these values were low in the higher quantiles compared to the reference group.

Fig 1 presents the smoothed distribution curve using the LMS method for the HAZ differences over time in males and females aged 0–36 months in selected years. A decreasing trend is seen in all of the percentile curves from 1992 to 2016, in which the levels are decreased in the higher percentile levels among both genders and in each age group. Differentiation of the plots noted that females' stunting percentile curves were marginally stable than those of male children. According to the findings stunting is more likely to occur in both males and females as they become older, However, when NFHS 1 curves were compared to the NFHS 4 curves, the NFHS 4 curves were found to be decreased.

## Discussion

Based on data obtained over a period of 24 years from 4 rounds of NFHS, the determinants associated with HAZ and trend in the HAZ distribution among children aged 0–36 months were identified using quantile regression. The LMS method was used to construct the curves of the 5th, 10th, 25th, 50th, 75th, 90th, and 95th percentiles. Research studies carried out around the

world have investigated the determinants of child nutrition status utilising distinctive econometric techniques. Most of the past investigations have generally utilized standard linear regression and logistic regressions. The utilisation of these regression estimation techniques may prompt misdirection of policy intervention measures, if the relationship between children's nutritional status and the different determinants are heterogeneous at the various percentiles [50]. One of the most important priority objectives in the agenda of the developing countries is to decrease child malnutrition. This study elucidates the use of quantile regression approach to investigate the effects of demographic, socio-economic and health-related factors on HAZ at different percentiles [18]. The conventional (linear and logistic) regression methods that were employed in earlier studies, but these approaches were limited to capture cross-distribution changes [3, 9, 26–30]. The analysis identified the child's age in months as one of the strongest determinants of the nutritional status. Results showed a negative increment in coefficients from lower percentiles to upper percentiles when compared with their first six months of life. The analysis also reveals stunting in the first six months of life and it increases with age [26, 51, 52].

The present study showed a negative association between the mother's educational status and HAZ. It was observed that the children of mothers with less education had worse HAZ scores than children of mothers with high education. This analysis suggests that mother's educational status plays a crucial role in child nutrition. Educated mothers have been shown to prioritize meeting the nutritional requirements of the children. They tend to access the available health facilities within their reach. Therefore, children of better-educated mothers appear to be well looked after as compared to 'not so well-educated mothers. Well-educated mothers were found to be capable enough to feed their children satisfactorily. They were shown to utilise the available health facilities in more optimum ways. Moreover, their health-related behavior's, including knowledge of nutrient-rich foods and practices were found to be better [53–55].

The present study showed a negative association and statistical significance between household economic status and HAZ. Children from the poorest, poorer, and middle household economic status had higher negative HAZ scores than the richest households. The economic condition of the family often influences the deprivation of infants. In this analysis, we find that among children with the lowest socioeconomic history the probabilities for being stunted, underweight and wasted were substantially greater. The families with lower incomes have a minimal supply of food, hence cannot ensure adequate food security, thus their growth and development get affected. Healthcare cannot meet their essential needs, influencing development and growth. It is observed that the children of low-income families had worse HAZ scores than children from high-income households. The study illustrates that factors like child age, birth order of the child, mother's education and household economic status were more important than aggregate economic conditions [56]. For the factor 'size at birth', it was found that those male children whose weight at birth was less than desired suffered a nutritional drawback in height as compared to their peers.

The mother's nutritional status (BMI) was also associated and statistical significance with children's nutrition. The results showed a substantial effect on a mother's BMI on a child's nutritional status. Children of mothers with lower BMI are more likely to be less HAZ when compared with their counterparts.

## Conclusion

This quantile regression suggests that the effects of different covariates worked differently across the HAZ distribution. Predictors of stunting viz: child's age in months, birth order, size at birth and wealth index and their associations vary across different quantiles of stunting. Since India's socio-economic and demographic characteristics have underlined the significant

disparities in many aspects, national strategies to tackle stunting should be tailored appropriately for various segments. It is suggested to carefully integrate applicable interventions according to the objective and target population for individuals' wellbeing and the country's development. Various measures could effectively combat child nutrition among lower socioeconomic strata. Increased awareness among family members on the right feeding practices using mass media or other community-based programs, subsidising the cost of nutrition-rich foods and ensuring adequate universal health care coverage.

## Strength and limitations of the study

The study depicts the trends in child nutritional status from 1992 to 2016 using the four rounds of NFHS data covering most parts of India. We utilised unweighted cases data from the NFHS survey, since quantile regression is the tool to use for these types of data without regard to survey design weights. As a default, QR employs a combination of the rank and resampling methods, as well as bootstrap to produce confidence intervals and standard errors for more robust statistical analysis.

The combination of quantile regression and the LMS approach is a more effective approach for finding the heterogeneous effect of demographic, socioeconomic and health factors on child Height for Age (HAZ) status.

Various intervention measures are required to consider the different effects of child's nutritional status determinants with various percentiles of the HAZ distribution. Child malnutrition can only be tackled with a multifaceted approach that includes targeted intervention, and it cannot be combated with a one-size-fits-all policy.

LMS curve with different percentiles for 1992(NFHS-1) and 2016(NFHS-4) quantified the trends in the performance of each gender and child's nutritional status. Like other studies of similar nature limitations of the analysis of the secondary data also apply to this study. Cross sectional data was used for this study.

## Acknowledgments

The authors gratefully acknowledge the National Institute of Nutrition (NIN), Indian Council of Medical Research, Hyderabad, India, for carrying out the study. The authors wish to thank Dr. R Hemalatha (Director, NIN) for her encouragement. The authors would also like to thank Dr. A. Laxmaiah Scientist G & HOD, Public Health Nutrition Division, NIN. The authors would also like to thank, Mounika Kandukuri (DST-INSPIRE Fellow) from Department of Statistics, Osmania University and both PLOS ONE reviewers whose reviews greatly improved the manuscript.

## Author Contributions

**Conceptualization:** Thirupathi Reddy Mokalla, Vishnu Vardhana Rao Mendu.

**Data curation:** Thirupathi Reddy Mokalla.

**Formal analysis:** Thirupathi Reddy Mokalla.

**Funding acquisition:** Vishnu Vardhana Rao Mendu.

**Methodology:** Vishnu Vardhana Rao Mendu.

**Supervision:** Vishnu Vardhana Rao Mendu.

**Writing – original draft:** Thirupathi Reddy Mokalla.

**Writing – review & editing:** Vishnu Vardhana Rao Mendu.

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
