## [Decision Letter · Decision Letter 0]

18 May 2021

PONE-D-21-11573

Application of Quantile Regression to examine changes in the distribution of Height for Age (HAZ) of Indian children aged 0-36 months using four rounds of NFHS data.

PLOS ONE

Dear Dr. Rao,

Thank you for submitting your manuscript to PLOS ONE. After careful consideration, we feel that it has merit but does not fully meet PLOS ONE’s publication criteria as it currently stands. Therefore, we invite you to submit a revised version of the manuscript that addresses the points raised during the review process.

The reviewers have raised important problems that related to the validity of the analysis. In particular, the current analysis does not address the issue of accounting for the sampling design in the analysis. 

We look forward to receiving your revised manuscript.

Kind regards,

Emanuele Giorgi

Academic Editor

PLOS ONE

Journal Requirements:

3. Please amend the manuscript submission data (via Edit Submission) to include author Thirupathi Reddy Mokalla.

Reviewers' comments:

Reviewer's Responses to Questions

**Comments to the Author**

1. Is the manuscript technically sound, and do the data support the conclusions?

Reviewer #1: Yes

Reviewer #2: Partly

2. Has the statistical analysis been performed appropriately and rigorously? 

Reviewer #1: Yes

Reviewer #2: Yes

3. Have the authors made all data underlying the findings in their manuscript fully available?

Reviewer #1: Yes

Reviewer #2: No

4. Is the manuscript presented in an intelligible fashion and written in standard English?

Reviewer #1: Yes

Reviewer #2: Yes

5. Review Comments to the Author

Reviewer #1: The authors of "Application of Quantile Regression to examine changes in the distribution of Height for

Age (HAZ) of Indian children aged 0-36 months using four rounds of NFHS data" submit an analysis that aimed to provide evidence of their objective, viz. "The objective of the study was to examine how the different determinants were heterogeneous in various percentiles of height for age (HAZ) distribution." They aptly lay out the background, methods, results and conclusions.

There are a number items that need to be addressed for which a de-identified file is attached that provides more comprehensive comments for the authors to improve the manuscript. Overarching in all of those is the following:

1. As the data are from a complex government survey, there are typically survey weights (aka design weights and post-stratification weights) applied to the data to account for survey design (design weights) or to adjust estimates to be generalizable to the population (post-stratification weights which will also contain the design weights). I do not see a discussion of these in the manuscript. If the authors are generalizing to the population, then post-stratification weights should be applied, otherwise, two things may happen:

1a. If post-stratification weights are not applied, results are not generalizable to the population;

1b. The design weights are meant to correct for the fact that complex surveys of this type are not a simple random sample so that the observations are not independent and identically distributed (IID) so that statistical procedures that rely on this fact (viz. normal distributions) will not give correct estimates. The saving grace here is that the beauty of quantile regression does not require IID and so this is a good reason to use quantile regression in this case [quantile regression has its origins to model data where unequal variance exists (Koenker, 2005)].

Reviewer #2: 1. All background covers the stunting among under-five children, however under-three data was used in the analysis. It should be revised and prevalence and determinants for under-three data will be presented.

2. Outcome is a composite one and age was included in HAZ definition. How was it also included in the model as an independent variable?

3. How “size at the time of birth (small, average and large)” was defined in this study?

4. Please mention the version of SAS software in the method section.

5. Please mention the level of statistical significance in the method section.

6. Please add explanation about the approach for variable selection in the modelling process.

7. Based on the comment 2, please clarify this interpretation “If the child's age increases, the coefficient values increase and these values are low in the lower quantiles compared to the upper quantiles.”

8. Confidence intervals for the effects should be presented.

9. Are presented effects in tables univariate or multivariable?

10. Please mention how many babies were included in each percentile of response.

11. The manuscript requires grammatical and linguistic editing.

6. PLOS authors have the option to publish the peer review history of their article (what does this mean?). If published, this will include your full peer review and any attached files.

Reviewer #1: No

Reviewer #2: No

---

## [Author Response · Author response to Decision Letter 0]

8 Jul 2021

Journal Requirements Author Response

 Thank you for highlighting the inconsistencies in the style requirements. We have now amended the manuscript to align with these guidelines.

 All the supporting data are available in the manuscript. We used secondary data “NFHS-1,2,3 and 4” for this study, which is publicly accessible at 

https://dhsprogram.com/what-we-do/survey/survey-display-355.cfm.

3. Please amend the manuscript submission data (via Edit Submission) to include author Thirupathi Reddy Mokalla.

 Thank you. We have done the same

 Thank you for highlighting the inconsistencies. We have now modified the abstract and it is identical at both the places

 The study employed data from four rounds of publicly accessible National Family Health Survey (NFHS) data. NFHS has obtained ethical clearance before administering the surveys. Ethical approval is not needed for this study. The same is mentioned in page - 17.

Reviewer 1 Comments

Reviewer Comments Author Response

Reviewer #1: The authors of "Application of Quantile Regression to examine changes in the distribution of Height for Age (HAZ) of Indian children aged 0-36 months using four rounds of NFHS data" submit an analysis that aimed to provide evidence of their objective, viz. "The objective of the study was to examine how the different determinants were heterogeneous in various percentiles of height for age (HAZ) distribution." They aptly lay out the background, methods, results and conclusions.

There are a number items that need to be addressed for which a de-identified file is attached that provides more comprehensive comments for the authors to improve the manuscript. Overarching in all of those is the following:

As the data are from a complex government survey, there are typically survey weights (aka design weights and post-stratification weights) applied to the data to account for survey design (design weights) or to adjust estimates to be generalizable to the population (post-stratification weights which will also contain the design weights). I do not see a discussion of these in the manuscript. If the authors are generalizing to the population, then post-stratification weights should be applied, otherwise, two things may happen:

1a. If post-stratification weights are not applied, results are not generalizable to the population;

1b. The design weights are meant to correct for the fact that complex surveys of this type are not a simple random sample. so that the observations are not independent and identically distributed (IID) so that statistical procedures that rely on this fact (viz. normal distributions) will not give correct estimates. 

The saving grace here is that the beauty of quantile regression does not require IID and so this is a good reason to use quantile regression in this case [quantile regression has its origins to model data where unequal variance exists (Koenker, 2005)]. 

 1a. Stratification weights are not used in computing the quantile regression estimates. 

Belasco, E.J., B. Chidmi, mentioned study results indicated that the two are not statistically different when using quantile regression, meaning both are consistent and implying the unweighted quantile regression is preferred because it is more efficient. 

https://scholarworks.montana.edu/xmlui/handle/1/8976

1b. We thank the reviewer for the comment and we are completely in alignment with the reviewer’s view on quantile regression analysis which does not need the design weights because it does not follow the Gaussian/ Normal Distribution. 

http://www.econ.uiuc.edu/~roger/research/intro/rq.pdf

Reviewer 2 Comments

Reviewer Comments Author Response

1. All background covers the stunting among under-five children, however under-three data was used in the analysis. It should be revised and prevalence and determinants for under-three data will be presented. 

 According to the National Family Health Survey (NFHS-1,2,3 and 4), the prevalence of stunted among children 0-36 months was 44.6 %, 45.5 %, 41.4%, and 36.6 %, respectively and the same were included in the page number 15. 

Introduction and background were changed accordingly 

2. Outcome is a composite one and age was included in HAZ definition. How was it also included in the model as an independent variable?

 The age of the child has a strong non-linear impact on his/her nutritional status. To capture this effect, we included dummy variables that reflect different age categories. 

https://pubmed.ncbi.nlm.nih.gov/16087414/

3. How “size at the time of birth (small, average and large)” was defined in this study?

 Since birth weight may not be known for

many babies in the field condition, the mother’s perception of the

baby’s size at birth was obtained for all births and classified size of the child at birth by NFHS questionnaire as follows: Very Large, Larger than Average, 

Average, Smaller than Average and Very Small.

However, we have collapsed it into three categories which are 

Large (containing Very large and Larger than Average), 

Average, and Small (containing Smaller than Average and Very Small).

4. Please mention the version of SAS software in the method section.

 This analysis was performed using the SAS University Edition software and SAS® OnDemand for Academics. Same thing was included in the method section page number 19.

https://www.sas.com/en_us/software/university-edition.html

5. Please mention the level of statistical significance in the method section.

 Statistically significant values p< 0.01, p<0.05, and p< 0.1 were include in the revised manuscript (page number 18).

6. Please add explanation about the approach for variable selection in the modelling process.

 Thank you for this suggestion. This explanation is now added in the manuscript. please see page number 18.

7. Based on the comment 2, please clarify this interpretation “If the child's age increases, the coefficient values increase and these values are low in the lower quantiles compared to the upper quantiles)

 Thanks for highlighting this we have corrected the typo graphical error in the page number 20

8. Confidence intervals for the effects should be presented.

 we have added the confidence intervals to the estimates in the tables, changes can be found from table 2 to 6

9. Are presented effects in tables univariate or multivariable?

 We have used the Multivariate quantile regression to estimate the coefficients on each percentile of the dependent variable (HAZ)

10. Please mention how many babies were included in each percentile of response.

 We have included total babies for each table (table 2 to 6).

We had mentioned the total no of children included for each table with gender separated values. because the quantile regression uses the full distribution for every quantile coefficient estimation. 

References 

1. http://www.econ.uiuc.edu/~roger/research/rq/rq.pdf

2. https://link.springer.com/article/10.1186%2Fs40536-017-0048-4

3. https://journals.sagepub.com/doi/10.1177/0013164419837321

11. The manuscript requires grammatical and linguistic editing. Thank you, we have edited and fixed the linguistic issues to the best of our ability.

---

## [Decision Letter · Decision Letter 1]

8 Feb 2022

PONE-D-21-11573R1Application of Quantile Regression to examine changes in the distribution of Height for Age (HAZ) of Indian children aged 0-36 months using four rounds of NFHS data.PLOS ONE

Dear Dr. Rao,

Thank you for submitting your manuscript to PLOS ONE. After careful consideration, we feel that it has merit but does not fully meet PLOS ONE’s publication criteria as it currently stands. Therefore, we invite you to submit a revised version of the manuscript that addresses the points raised during the review process. The current versions of the manuscript presents a quantile regression of data on HAZ scores, however, the justification for the use this methodology is not well justified.What is not clear is if the focus of this study is on the advantages of quantile regression over standard linear regression methods or if the focus is on the modelling of HAZ.Since I think the latter is more likely, then the paper has not illustrated and showed convincingly 1) what is the link between the research question and quantile regression and 2) why the research question cannot be answered using standard regression. I advise the authors to revise their manuscript substantially to expand more on the research question and the interpretational advantages of quantile regression. For example, why should risk factors have different effects on the quantiles of the HAZ distribution? The paper also does not include references to statistical models that have been used to analyze HAZ data and does not provide any explanation as to why those methods cannot be used for the present study.The revised version of the manuscript should therefore provide a more thorough review of previews statistical work in this field and comparisons should be carried out with the chosen method. Please submit your revised manuscript by Mar 25 2022 11:59PM. If you will need more time than this to complete your revisions, please reply to this message or contact the journal office at plosone@plos.org. Please include the following items when submitting your revised manuscript:A rebuttal letter that responds to each point raised by the academic editor and reviewer(s). You should upload this letter as a separate file labeled 'Response to Reviewers'.A marked-up copy of your manuscript that highlights changes made to the original version. You should upload this as a separate file labeled 'Revised Manuscript with Track Changes'.An unmarked version of your revised paper without tracked changes. You should upload this as a separate file labeled 'Manuscript'.

We look forward to receiving your revised manuscript.

Kind regards,

Emanuele Giorgi

Academic Editor

PLOS ONE

Reviewers' comments:

Reviewer's Responses to Questions

**Comments to the Author**

1. If the authors have adequately addressed your comments raised in a previous round of review and you feel that this manuscript is now acceptable for publication, you may indicate that here to bypass the “Comments to the Author” section, enter your conflict of interest statement in the “Confidential to Editor” section, and submit your "Accept" recommendation.

Reviewer #1: All comments have been addressed

2. Is the manuscript technically sound, and do the data support the conclusions?

Reviewer #1: Yes

3. Has the statistical analysis been performed appropriately and rigorously? 

Reviewer #1: Yes

4. Have the authors made all data underlying the findings in their manuscript fully available?

Reviewer #1: Yes

5. Is the manuscript presented in an intelligible fashion and written in standard English?

Reviewer #1: Yes

6. Review Comments to the Author

Reviewer #1: Thank you for making the requested changes. Please see my anonymized editorial review in Word with Track changes for grammar updates. This will improve the readability of the manuscript.

7. PLOS authors have the option to publish the peer review history of their article (what does this mean?). If published, this will include your full peer review and any attached files.

Reviewer #1: No

---

## [Author Response · Author response to Decision Letter 1]

6 Mar 2022

06th March,2022

Dear Dr. Emanuele Giorgi,

Academic Editor.

PLOS ONE

Dear Sir,

We were pleased to have an opportunity to revise our manuscript entitled " Application of Quantile Regression to examine changes in the distribution of Height for Age (HAZ) of Indian children aged 0-36 months using four rounds of NFHS data. (PONE-D-21-11573)" to PLOSONE. We appreciate the time and effort that you and the reviewers and editor have dedicated to providing your valuable feedback on our manuscript. We are grateful to the editor and reviewers for their insightful comments on our paper. In the revised manuscript, we have carefully considered editors comments and suggestions. As instructed, we have attempted to briefly explain changes made in reaction to all comments. We reply to each comment in point-by-point fashion. We have the color-coded revised manuscript as text.

The responses to the concerns raised by editors/reviewers are below 

The editors' comments were very helpful overall, and we are appreciative of such constructive feedback on our original submission. After addressing the issues raised, we feel the quality of the paper is much improved. 

Funding: No specific funding has been received for this study.

Thanking You

Sincerely,

Dr. M. Vishnu Vardhana Rao

DIRECTOR, ICMR-NIMS

Min. of Health and Family Welfare

Govt. of India, New Delhi

Responses to editor comments

1) Since I think the latter is more likely, then the paper has not illustrated and showed convincingly 1) what is the link between the research question and quantile regression and

Response: The research question of the current study is to examine how the different determinants were heterogeneous in various percentiles of height for age (HAZ) distribution. In this study we explore the risk factors for childhood malnutrition in India using the quantile regression. 

Traditional regression modelling approaches (linear/logistic regression) used to estimate the risk factors of childhood stunting often tend to oversimplify the complex interplay of a battery of risk factors through emphasizing a solitary risk and intervention. Recent studies in public health and epidemiological research called upon the application of a systems approach that focuses on examining the determinants of stunting in its entirety and seek to avoid the possibility of incorrect estimation of risk factors and programmatic interventions owing to any oversimplification. In addition, recent studies have also noted that the impact of risk factors at the lower tail of the distribution could be considerably different contrary to the population mean or at the higher end of the distribution. Therefore, in any population with a considerable degree of nutritional transition and disproportionate burden of anthropometric failure, it is critical to recognize and address the risk factors of stunting at the different ends of the spectrum and meticulously identify the actual effect of the risk factors at a different part of the distribution (i.e. lower tail, mean and upper bound of the distribution of z-score) to design and implement appropriate interventions. Quantile regression model quantiles of the outcome as a function of covariates and provides an opportunity to examine whether covariates have differential effects across the z-score distribution, particularly towards the lower tail. On the other hand, previous studies have largely examined the risk factors of stunting by applying logistic regression models for dichotomized versions of the Z-score (e.g. stunted vs. not stunted) or linear regression models for the continuous Z-score.

 2) why the research question cannot be answered using standard regression. I advise the authors to revise their manuscript substantially to expand more on the research question and the interpretational advantages of quantile regression.

 For example, why should risk factors have different effects on the quantiles of the HAZ distribution? 

Response: We wish to study the different independent variables effect on the HAZ (dependent variable). In the linear regression we wish to study the mean affect on the dependent variable with the independent variables. Various studies have been identified in the prior literature, conducted in a wide range of developing nations and India, to explore the risk factors for child malnutrition by using a variety of methodologies and regression models. The linear regression model, multilevel or ordinal logistic regression models, and multilevel or ordinal logistic regression models were all used to analyse the nutritional status of children in the majority of the articles.

here we used the quantile regression over the standard regression (logistic regression and other methods), using the standard regression/ other regression we get one coefficient value and with their independent values only we study. The majority of the prior studies have been conducted with the HAZ score being dichotomized or categorise using logistic regression or linear regression, respectively. One significant drawback is that the dependent variable must be categorised according to certain criteria, which may reduce the statistical power of the odds ratio by a significant margin. When applied to a continuous response variable and a set of predictors, the linear regression (LR) model determines the association between them by using a conditional mean function. The LR technique, on the other hand, cannot be extended to non-central (other than mean) locations, and as a result, the complete conditional distribution of the outcome variable cannot be investigated. In practise, the model assumptions might be broken, and as a consequence, the estimated model may not accurately reflect the fluctuation of the response distribution in the real world.

But with the quantile regression we wish to study the HAZ effect on the different quaintiles (like 5,10, 25,70,75,90 and 95), with this we get the various coefficient value with respect to the quaintile , by using these values we can study the effect of the each independent variable on the different quiantile with independent variables. 

Quantile regression model quantiles of the outcome as a function of covariates and provides an opportunity to examine whether covariates have differential effects across the z-score distribution, particularly towards the lower tail. On the other hand, previous studies have largely examined the risk factors of stunting by applying logistic regression models for dichotomized versions of the Z-score (e.g. stunted vs. not stunted) or linear regression models for the continuous Z-score.

3) The paper also does not include references to statistical models that have been used to analyze HAZ data and does not provide any explanation as to why those methods cannot be used for the present study.

The revised version of the manuscript should therefore provide a more thorough review of previews statistical work in this field and comparisons should be carried out with the chosen method.

Response: In manuscript we mentioned at page number 4 and reference numbers 26-32. 

And also, below references were included in the manuscript 

Reference:

1. Fenske, N., Kneib, T., Hothorn, T. (2011) Identifying Risk Factors for Severe Childhood Malnutrition by Boosting Additive Quantile Regression, Journal of the American Statistical Association, 106:494, 494-510, DOI: 10.1198/jasa.2011.ap09272 

2. Fenske, N., Burns, J., Hothorn, T., Rehfuess, E.A. (2013) Understanding Child Stunting in India: A Comprehensive Analysis of Socio-Economic, Nutritional and Environmental Determinants Using Additive Quantile Regression. PLoS ONE 8(11): e78692. https://doi.org/10.1371/journal.pone.0078692

3. Banerjee K, Dwivedi LK (2020) Disparity in childhood stunting in India: Relative importance of community-level nutrition and sanitary practices. PLoS ONE 15(9): e0238364. https://doi.org/10.1371/journal.pone.0238364

---

## [Editor Report · Decision Letter 2]

10 Mar 2022

Application of Quantile Regression to examine changes in the distribution of Height for Age (HAZ) of Indian children aged 0-36 months using four rounds of NFHS data.

PONE-D-21-11573R2

Dear Dr. Rao,

We’re pleased to inform you that your manuscript has been judged scientifically suitable for publication and will be formally accepted for publication once it meets all outstanding technical requirements.

Kind regards,

Emanuele Giorgi

Academic Editor

PLOS ONE
---

## [Editor Report · Acceptance letter]

19 May 2022

PONE-D-21-11573R2 

Application of Quantile Regression to examine changes in the distribution of Height for Age (HAZ) of Indian children aged 0-36 months using four rounds of NFHS data. 

Dear Dr. Rao Mendu:

I'm pleased to inform you that your manuscript has been deemed suitable for publication in PLOS ONE. Congratulations! Your manuscript is now with our production department. 

Kind regards, 

on behalf of

Dr. Emanuele Giorgi 

Academic Editor

PLOS ONE